# Context-dependent deposition and regulation of mRNAs in P-bodies

**Congwei Wang[1], Fabian Schmich[2,3], Sumana Srivatsa[2,3], Julie Weidner[1], Niko Beerenwinkel[2,3], Anne Spang[1]***

[1]Growth and Development, Biozentrum, University of Basel, Basel, Switzerland; [2]Department of Biosystems Science and Engineering, ETH Zürich, Basel, Switzerland; [3]Swiss Institute of Bioinformatics, Basel, Switzerland

**Abstract** Cells respond to stress by remodeling their transcriptome through transcription and degradation. Xrn1p-dependent degradation in P-bodies is the most prevalent decay pathway, yet, P-bodies may facilitate not only decay, but also act as a storage compartment. However, which and how mRNAs are selected into different degradation pathways and what determines the fate of any given mRNA in P-bodies remain largely unknown. We devised a new method to identify both common and stress-specific mRNA subsets associated with P-bodies. mRNAs targeted for degradation to P-bodies, decayed with different kinetics. Moreover, the localization of a specific set of mRNAs to P-bodies under glucose deprivation was obligatory to prevent decay. Depending on its client mRNA, the RNA-binding protein Puf5p either promoted or inhibited decay. Furthermore, the Puf5p-dependent storage of a subset of mRNAs in P-bodies under glucose starvation may be beneficial with respect to chronological lifespan.
DOI: https://doi.org/10.7554/eLife.29815.001

*For correspondence:
anne.spang@unibas.ch

**Competing interests:** The authors declare that no competing interests exist.

## Introduction

Cells are often subjected to environmental fluctuations, such as nutrient deficiency, osmotic shock and temperature change. Therefore, cells have evolved a variety of cellular mechanisms to adapt and survive under those conditions, which are generally referred to as stress responses (*Mager and Ferreira, 1993*). Regulation of transport, translation and stability of mRNAs are among the first acute responses contributing to the rapid adjustment of the proteome. In response to stress, protein synthesis is globally attenuated, but a subset of mRNAs, necessary to cope with the stress, is still subject to efficient translation (*Ashe et al., 2000*). Non-translating mRNAs are mostly deposited into processing bodies (P-bodies) and stress granules (SGs), which are two types of ribonucleoprotein particles (RNPs), conserved from yeast to mammals. As the formation of both granules is induced under diverse stress conditions and a number of components appear to be shared, their precise role in stress response is still a matter of debate (*Kulkarni et al., 2010*; *Mitchell et al., 2013*). While P-bodies and SGs both participate in repression of translation and mRNA storage, P-bodies represent also the main site for mRNA degradation via the 5'-decapping-dependent pathway, the 5'–3' exonuclease Xrn1p and transport (*Decker and Parker, 2012*; *Davidson et al., 2016*). In addition to the decay occurring in P-bodies, a 3'–5' exonucleolytic pathway, via the exosome, exists (*Anderson and Parker, 1998*). More recently, a co-translational RNA decay pathway has been discovered, which responds to ribosome transit rates (*Pelechano et al., 2015*; *Sweet et al., 2012*). Interestingly, some of the P-body components such as the helicase Dhh1p and the exonuclease Xrn1p also act in the co-translational pathway. Moreover, other P-body components such as the decapping activator Dcp2p have been found to associate with polysomes (*Weidner et al., 2014*). How and which mRNAs are selected into the different pathways, in particular under stress, remains

elusive, partly because unbiased methods to identify RNA species are still not widely used. Here, we devised a novel method to identify RNA species in RNPs, in particular P-bodies.

The protein composition of P-bodies has been extensively studied in both yeast and metazoans (*Kulkarni et al., 2010*), yet, numerous auxiliary and transient components are still being discovered (*Hey et al., 2012*; *Ling et al., 2014*; *Weidner et al., 2014*), suggesting a tight regulation of the RNA inventory and fate. Very little is known, however, about the regulation of mRNA fate in P-bodies. To date, the RNA inventory in P-bodies under a particular stress remains unclear, and in yeast only a handful of mRNAs have been confirmed to localize to P-bodies (*Brengues et al., 2005*; *Cai and Futcher, 2013*; *Lavut and Raveh, 2012*). Several studies have proposed P-bodies to act not only as decay compartments but also to store and later release RNAs back into the translation pool, particularly upon stress removal. This notion is primarily supported by an observed dynamic equilibrium of mRNA localization between polysomes and P-bodies (*Brengues et al., 2005*; *Kedersha et al., 2005*; *Teixeira et al., 2005*). Recently, this model has been challenged and it was proposed that Xrn1p-dependent decay might occur outside P-bodies (*Sweet et al., 2012*), which is supported by findings that the 5′ decapping machinery is present at membrane-associated polysomes under non-stress conditions (*Huch et al., 2016*; *Weidner et al., 2014*). Still, a prevailing hypothesis in the field is that specific mRNAs preferentially accumulate in P-bodies under different stresses promoting cell adaption and survival (*Decker and Parker, 2012*). In support of this concept, the number, morphology and half-life of P-bodies vary depending on the particular stress. For example, under glucose starvation only a few, large, long-lived P-bodies are observed microscopically, whereas $Ca^{2+}$ stress produces numerous, small P-bodies that disappear within 30 to 45 min after the initial induction (*Kilchert et al., 2010*). Lacking a global picture of mRNA species in P-bodies greatly hinders the study of the functional role of P-bodies in mRNA turnover and stress response.

A major obstacle in the universal identification of mRNAs present in P-bodies is that at least a portion of the transcripts are likely engaged in deadenylation or degradation, and, hence, commonly used oligo(dT) purification provides an incomplete and biased picture of mRNAs present in P-bodies. We overcame this obstacle by adapting and improving a crosslinking affinity purification protocol (*Weidner et al., 2014*) to globally isolate P-body associated transcripts. We demonstrate that P-bodies contain distinct mRNA species in response to specific stresses. The sequestered transcripts underwent different fates depending on their function, for example: mRNAs involved in overcoming stress were stabilized while others were degraded. Similarly, mRNA decay kinetics differed depending on the mRNA examined. Our observations are consistent with a dual role of P-bodies in mRNA degradation and storage. Under glucose starvation, the RNA-binding protein Puf5p plays a central role in regulating the decay of a set of mRNAs and is also responsible for the localization and stability of another set. Moreover, the stabilization of at least one mRNA in a Puf5-dependent manner may contribute to chronological lifespan.

## Results

### A novel method to isolate RNAs sequestered into P-bodies

To determine the mRNA species sequestered into P-bodies upon different stress conditions, we combined and improved a method based on in vivo chemical crosslinking and affinity purification, which we had previously used to identify regulators and protein components of P-bodies (*Weidner et al., 2014*) with commonly used techniques to generate RNA libraries for subsequent RNA-Seq (*Hafner et al., 2010*; *Kishore et al., 2011*) (*Figure 1A*). We refer to this method as chemical Cross-Linking coupled to Affinity Purification (cCLAP). Our earlier work showed that P-bodies in yeast are in very close proximity to the endoplasmic reticulum (ER) and that they fractionate with ER membranes (*Kilchert et al., 2010*; *Weidner et al., 2014*). To explore the mRNA content of P-bodies, either Dcp2p or Scd6p, which are part of the 5′ and the 3′UTR-associated complex of P-bodies, respectively, were chromosomally tagged with a $His_6$-biotinylation sequence-$His_6$ tandem tag (HBH) (*Tagwerker et al., 2006*; *Weidner et al., 2014*). P-bodies were either induced through glucose starvation or through the addition of $CaCl_2$ or NaCl. We chose $CaCl_2$ as stressor because secretory pathway mutants induce P-bodies through a $Ca^{2+}$/calmodulin-dependent pathway, which is mimicked by the addition of $Ca^{2+}$ to the medium (*Kilchert et al., 2010*). Notably, this induction pathway is different from the one employed by the cell upon glucose starvation. NaCl was selected as an alternative

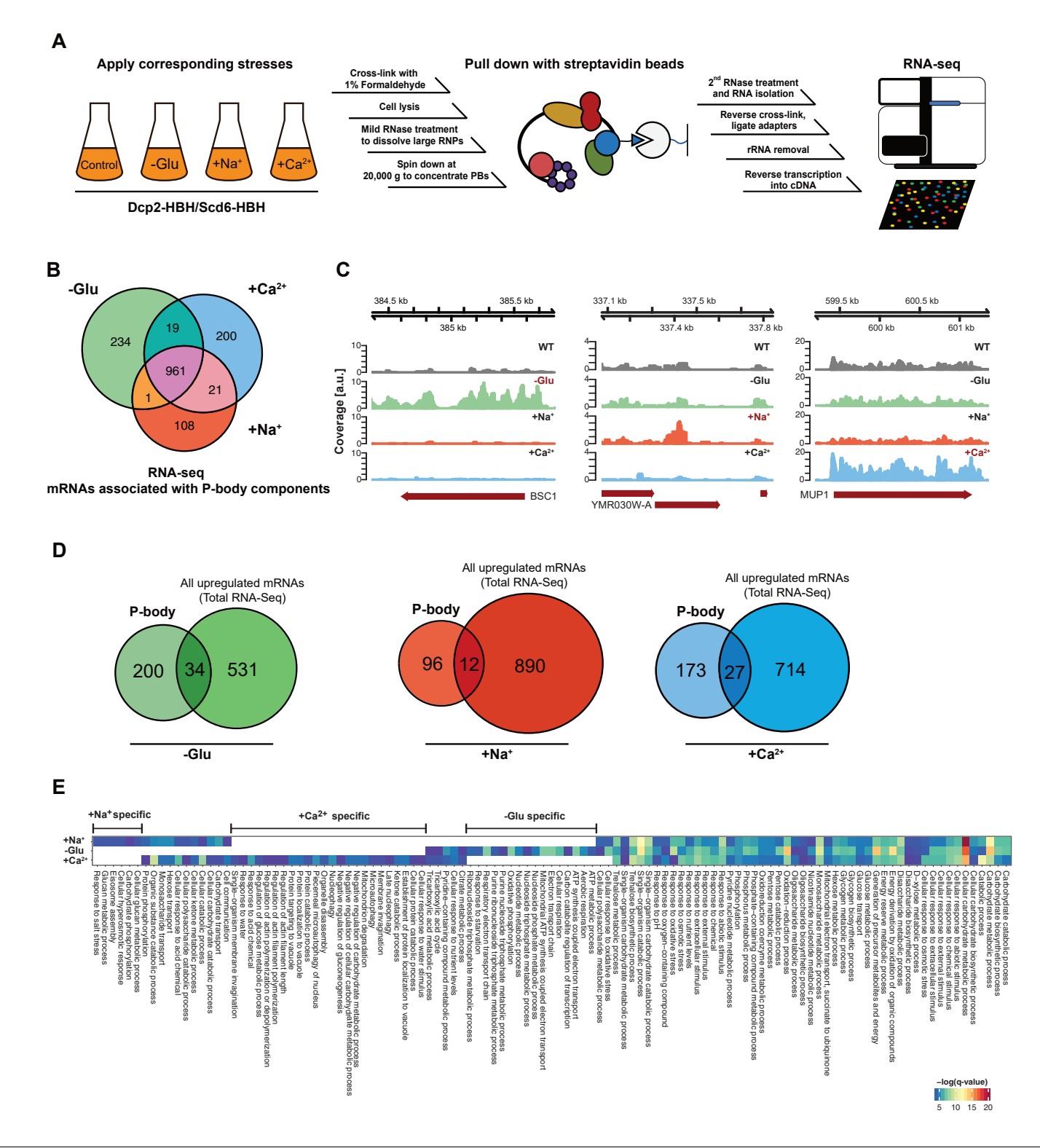

**Figure 1.** RNA-Seq reveals stress-specific mRNA subsets associated with P-body components at membranes. (**A**) RNA-Seq library preparation workflow. Cells expressing Dcp2-HBH or Scd6-HBH were stressed for 10 min, followed by cross-linking with formaldehyde. After cell lysis, centrifugation was performed to enrich membrane fractions. Cross-linked complexes were subsequently purified via streptavidin affinity purification. mRNAs were isolated and ligated with adapters. cDNA libraries were prepared by reverse transcription and sequenced using single-read RNA-Seq. (**B**) Venn diagram illustrating the intersections among mRNAs associated with P-body components ($p < 0.05$) under glucose depletion and osmotic stress conditions with

*Figure 1 continued on next page*

*Figure 1 continued*

Na$^+$ or Ca$^{2+}$, relative to the no stress condition as determined by RNA-Seq. (**C**) Read coverage plots (average over five biological replicates) of RNA-Seq data mapped to P-body associated genes under specific stress conditions. (**D**) Venn diagrams showing the intersections between mRNAs specifically associated with P-body components under glucose depletion, Na$^+$ or Ca$^{2+}$ stresses and all mRNAs that are upregulated upon the same treatment according to total RNA-Seq. (**E**) Enrichment analysis of P-body-associated genes under different stress conditions against Gene Ontology's (GO) biological processes (BP). Significantly enriched pathways (q-value <0.05) from hypergeometric tests are presented in a clustered heatmap. Rows and columns correspond to stress conditions and pathways, respectively, and the negative logarithms of q-values are color-coded from blue (low) to red (high).

DOI: https://doi.org/10.7554/eLife.29815.002

The following figure supplements are available for figure 1:

**Figure supplement 1.** Reproducibility of datasets derived from RNA-Seq and Total RNA-Seq.

DOI: https://doi.org/10.7554/eLife.29815.003

**Figure supplement 2.** Flow chart of the analysis of RNA-Seq data.

DOI: https://doi.org/10.7554/eLife.29815.004

hyperosmotic stress to determine whether different hyperosmotic stresses would elicit the same or different responses. We chose formaldehyde as cross-linking agent because it can be directly applied to the culture medium and is easily and rapidly quenchable allowing precise cross-linking conditions without introducing any unwanted stress like through centrifugation or medium changes prior to the cross-link reaction. Yeast cells were exposed to each stressor for 10 min, cross-linked and, after lysis, P-bodies were enriched from the membrane fraction through the HBH-tag present on either Dcp2p or Scd6p. Strictly speaking we are enriching mRNAs that can be crosslinked to Dcp2p or Scd6p or their interaction partners under stress conditions. Given that we identified P-body components previously using this method (*Weidner et al., 2014*), and that the Scd6p experiment clustered well with the ones performed with Dcp2p, makes it likely that the RNAs, we identified are present in P-bodies. We chose to stress the cells for only 10 min in order to exclude any contribution of SGs, which are not present at this time point (*Kilchert et al., 2010*) and *Figure 1—figure supplement 1A*). Libraries for RNA-Seq were prepared in two ways: either using PAGE purification with radiolabeled mRNAs or using a column-based purification method (*Supplementary file 1*).

Principal Component Analysis (PCA) performed on the read count profile for each condition from the aligned RNA-Seq data of the five independent biological replicates generated four clusters, corresponding perfectly to the three stress conditions plus the unstressed control (*Figure 1—figure supplement 1B*). Similarly, stress and control conditions clustered well using pair-wise correlation analyses (*Figure 1—figure supplement 1C*). Neither the tagged P-body component nor the purification method used for RNA-Seq sample preparation perturbs the clustering pattern, indicating a high degree of reproducibility of our method. Given that we used two types of hyperosmotic stress, it is not surprising that the Ca$^{2+}$ and Na$^+$ datasets cluster more closely than the ones derived from glucose starvation conditions. Yet, being able to detect differences between the two osmotic shock conditions further exemplifies the robustness of our approach. Therefore, cCLAP is a valid method to determine the RNA content of RNPs.

## The nature of P-body sequestered RNAs is stress-dependent

In total, we identified 1544 mRNAs statistically significantly associated with P-bodies under glucose depletion and Na$^+$ and Ca$^{2+}$ stresses, relative to the unstressed condition (*Figure 1B*, *Figure 1—figure supplement 2* and *Supplementary file 2*). While about 65% of the detected mRNAs were common between stresses, approximately 35% of the RNAs were specific to an individual stress (*Figure 1B*). Reads on stress-specific targets were distributed over the entire length without any preferential accumulation or depletion at the 5' or 3' UTRs as exemplified by the selected transcripts (*Figure 1C*).

To ensure the specificity of the mRNAs associated with P-body components, we performed RNA-Seq experiments on the total RNA content under control as well as the different stress conditions (*Figure 1—figure supplement 1D–F*, *Supplementary file 3*). mRNAs that were upregulated upon any of the stresses, as determined by total RNA-Seq, were by and large not enriched in the corresponding fraction of the P-body components with less than 15% overlap between the RNAs

generally upregulated in stress response and the RNAs pulled down by P-body components (*Figure 1D*). Moreover, from the glucose-starvation P-body component associated mRNA pool, polysome-associated mRNAs identified under the same stress (*Arribere et al., 2011*) were eliminated. These data then allowed us to determine whether a general feature such as gene length would contribute to the likelihood to be associated with P-bodies under a particular stress. Therefore, we plotted the length of the genes associated with P-body components and of genes generally upregulated under specific stress conditions (*Figure 1—figure supplement 1G*). P-body component associated mRNAs were shorter under glucose starvation and longer under osmotic stress than the generally upregulated mRNAs under the respective stress conditions. Thus, gene length may provide a bias to whether or not its gene product is associated with P-body components under specific stress conditions.

If mRNA deposition in P-bodies was context-dependent, one would expect an enrichment of mRNAs belonging to the same pathways/processes. To test this notion, we employed Gene Ontology (GO) enrichment analysis (biological process) (*Figure 1E*). Consistent with the Venn diagram (*Figure 1B*), a number of biological processes were shared by all three stress conditions, yet many GO terms were specific to one particular stress, suggesting that association of mRNAs with P-body components is, in general, context-dependent. For example, within the glucose specific set, we found a group of processes related to mitochondrial oxidative phosphorylation (herein referred to as mitochondria-related mRNAs). This group is of particular interest, as mitochondria respiration genes are generally up-regulated upon glucose starvation (*Wu et al., 2004*). Taken together, our data suggest that a subset of mRNAs is sequestered in P-bodies in a stress-dependent manner.

## mRNAs localize to P-bodies in a context-dependent manner

Thus far, we have shown that mRNAs can be cross-linked to P-body components in a stress-dependent manner. To demonstrate that these mRNAs indeed localize to P-bodies, we employed fluorescence in situ hybridization coupled to immunofluorescence (FISH-IF; *Figure 2A*). We used Dcp2p as P-body marker for immunofluorescence. Since P-bodies exhibit a compact, dense structure (*Souquere et al., 2009*), the traditionally employed long probes (up to 1000 nt) used in FISH are not suitable for detection of mRNA in P-bodies. However, using multiple 50–100 nt FISH probes (4–8 per transcript) allowed us to detect specific mRNAs in P-bodies, as the no probe control only exhibited background staining (*Figure 2*, *Figure 2—figure supplement 1A*). Regardless, we may not be able to detect all mRNA molecules in the cell and are likely underestimating the extent of localization of mRNAs within P-bodies. In addition, transcripts in yeast are often present in less than 10 copies per cell (*Zenklusen et al., 2008*), which may hinder detection by this method. Moreover, most mRNAs are degraded in P-bodies (*Sheth and Parker, 2003*), therefore any given mRNA may be detected in P-bodies at any given time. Finally, our FISH-IF method is based on enzymatic fluorescence development and hence does not provide single molecule resolution and is not quantitative with respect to number of RNA molecules per spot. Again, the accessibility by the probe and also by the enzyme is potentially better in the cytoplasm than in a compact assembly such as the P-body. Taken theses constraints into consideration, we set the threshold at ≥1.5 fold enrichment over control mRNAs to determine P-body association.

We selected a set of mRNAs from each stress condition and determined their subcellular localization. Upon glucose depletion, seven mRNAs including both non-mitochondria-related (Group I: *BSC1*, *TPI1*, *RLM1*) and mitochondria-related (Group II: *ATP11*, *ILM1*, *MRPL38*, *AIM2*) groups, based on the GO pathways, showed significant co-localization with P-bodies (*Figure 2B and C*, *Figure 2— source data 1*) relative to background (*Figure 2—figure supplement 1B,C*, *Figure 2—figure supplement 1—source data 1*). To validate that the mRNA localization to P-bodies is stress-specific, we repeated the FISH-IF under osmotic stresses for three mRNAs (*Figure 2D*). None of them was significantly associated with P-bodies under these stress conditions (*Figure 2E*, *Figure 2E*-source data 1). Similarly, we found mRNAs that were specifically associated with P-bodies under a unique osmotic condition but not under the other stresses (*Figure 2—figure supplement 1D and E*, *Figure 2—figure supplement 1—source data 2*). We conclude that at least a subset of mRNAs must be selected for -or spared from- transport to P-bodies in a context-dependent manner.

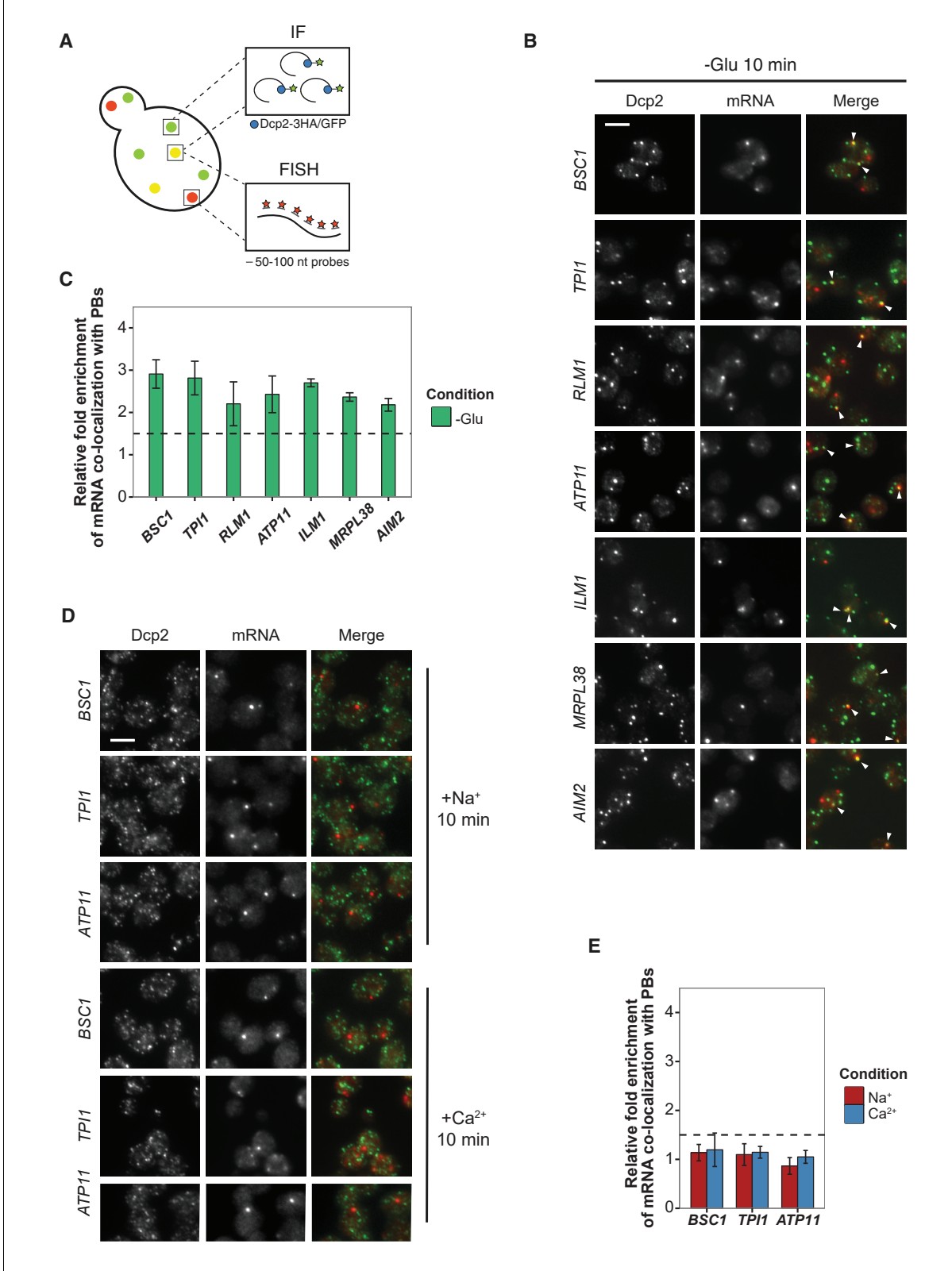

**Figure 2.** Validation of glucose-specific candidates by combined fluorescence in situ hybridization and immunofluorescence (FISH-IF). (**A**) Schematic representation of combined FISH-IF technique. Immunofluorescence staining was performed against P-body marker Dcp2 chromosomally tagged with 3 HA or GFP. To detect mRNAs accumulating in P-bodies, multiple short probes (50–100 nt) against the open reading frame (ORF) of each gene were used for FISH. (**B**) Fluorescence images of P-bodies and glucose-starvation-specific candidate mRNAs after glucose depletion. Cells expressing Dcp2-

*Figure 2 continued on next page*

*Figure 2 continued*

3HA were first grown in YPD media to mid-log phase and shifted to YP media lacking glucose for 10 min. Scale bar, 5 µm. Error bars, mean ±SEM. (**C**) Bar plot depicting the quantification of co-localization between candidate mRNAs and P-bodies. The percentage of co-localization was quantified as described in Materials and methods. The relative fold enrichment was subsequently calculated by normalizing the percentage of candidate mRNAs against the percentage of control mRNAs (*Figure 2—figure supplement 1C*). The dashed line represents an arbitrarily fixed threshold of 1.5 for determining significant P-body association. (**D**) Fluorescence images of P-bodies and glucose-specific candidate mRNAs under mild osmotic stress with $Na^+$ or $Ca^{2+}$. Cells expressing Dcp2-3HA were first grown in YPD media to mid-log phase and shifted to YPD media containing 0.5 M NaCl or 0.2 M $CaCl_2$ for 10 min. Scale bars, 5 µm. Error bars, mean ±SEM. (**E**) Same as (**C**) except stress conditions. Scale bar, 5 µm. Error bars, mean ±SEM.
DOI: https://doi.org/10.7554/eLife.29815.005

The following source data and figure supplements are available for figure 2:

**Source data 1.** Data used for plotting.
DOI: https://doi.org/10.7554/eLife.29815.007
**Source data 2.** Data used for plotting.
DOI: https://doi.org/10.7554/eLife.29815.008
**Figure supplement 1.** Evaluation of $Na^+$, $Ca^{2+}$ and non-candidate mRNAs by FISH-IF.
DOI: https://doi.org/10.7554/eLife.29815.006
**Figure Supplement 1—source data 1.** Data used for plotting.
DOI: https://doi.org/10.7554/eLife.29815.009
**Figure Supplement 1—source data 2.** Data used for plotting.
DOI: https://doi.org/10.7554/eLife.29815.010

## mRNAs experience divergent fates inside P-bodies

It has been proposed that mRNAs are not only decayed in P-bodies, but may be stored there and re-enter translation after stress subsides (*Brengues et al., 2005*). We found mRNAs that were potentially excellent candidates for being stored in P-bodies. The mitochondria-related genes were transcriptionally up-regulated following glucose starvation (*Figure 3—figure supplement 1B*), while at the same time transcripts were sequestered in P-bodies. To investigate the fate of P-body associated mRNAs further, we employed the 4TU non-invasive pulse-chase RNA labeling technique followed by qRT-PCR. With this technique, we can specifically label RNA before stress application and determine its decay rate (*Munchel et al., 2011*) (*Figure 3A*). To differentiate P-body specific degradation from the exosome decay pathway, we analyzed the mRNA half-life in the presence and absence of the P-body 5′−3′ exonuclease Xrn1p (*Figure 3B*). *ACT1* was used as endogenous reference gene due to its high stability during glucose starvation and because it was neither significantly associated with P-body components nor enriched in the total RNA samples under any stress condition tested (*Figure 3—figure supplement 1A*, *Figure 3—figure supplement 1—source data 1*, *Supplementary file 3*). No significant reduction in mRNA levels was observed for Group II mRNAs (*ATP11*, *ILM1*, *MRPL38* and *AIM2*) for up to one hour of glucose withdrawal, suggesting that those transcripts were stabilized inside P-bodies (*Figure 3B*, Group II). Consistently, after a rapid initial increase, the total transcript levels remained constant over the time course (*Figure 3—figure supplement 1B*, Group II). Conversely, the transcripts within group I (*BSC1*, *TPI1*, and *RLM1*) underwent Xrn1p-dependent decay (*Figure 3B*, *Figure 3—source data 1*, Group I). Intriguingly, the onset and the kinetic of the decay varied from mRNA to mRNA, indicating that individual intrinsic properties of the mRNAs may determine their half-lives within P-bodies. Likewise, the total mRNA levels were modulated in a similar way (*Figure 3—figure supplement 1B*, *Figure 3—figure supplement 1—source data 2*, Group I), hinting towards coordination between P-body specific decay and transcription. To ensure that the chase conditions were strong enough and to exclude any possible contribution from transcription, we repeated the pulse chase and added the maximal soluble concentration of uracil (32 mM). This uracil concentration did not change the outcome of our experiment (*Figure 3—figure supplement 1C*, *Figure 3—figure supplement 1—source data 3*). Since the concentration was only increased 1.6-fold, we turned to a transcriptional inhibitor, 1,10 phenanthroline, with the caveat that blocking transcription represents itself a significant stress. Yet, with the exception of *ILM1* mRNA, the fate of the mRNAs remained unchanged under glucose starvation (*Figure 3—figure supplement 1D*, *Figure 3—figure supplement 1—source data 4*), validating our pulse chase data with 4TU.

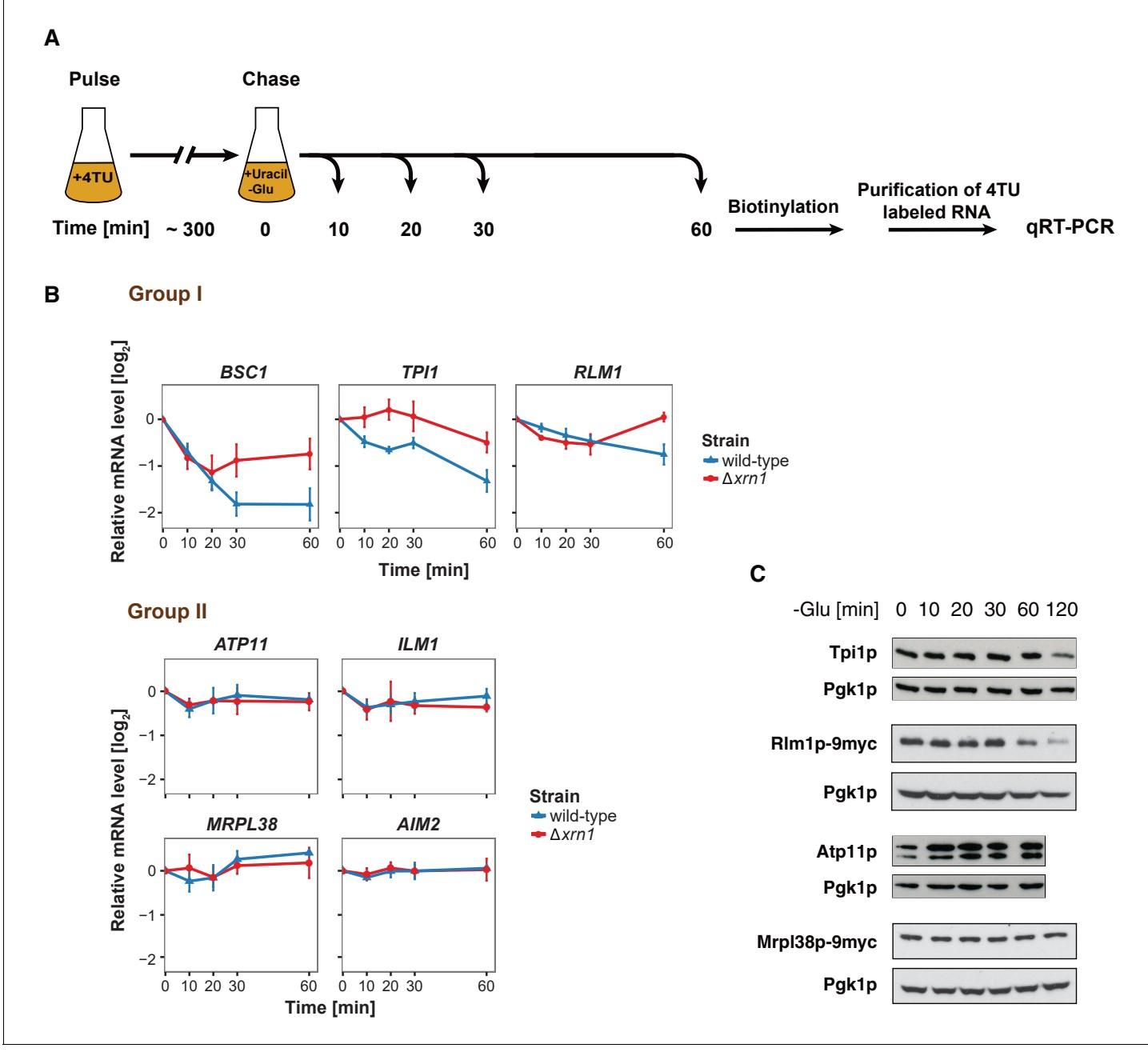

**Figure 3.** The stability of P-body associated mRNAs varies and can be categorized according to their GO terms. (A) Schematic illustration of pulse-chase protocol. Cells were grown in the presence of 0.2 mM 4-TU and shifted into media lacking glucose but containing 20 mM uracil. Cells were harvested at indicated time points after the shift. Total RNA was extracted and biotinylated. 4TU labeled RNA was purified and subsequently analyzed by qRT-PCR. (B) The stability of 4TU labeled candidate mRNAs was determined by qRT-PCR in wild type and Δxrn1 strains at indicated time points following a shift to glucose-depleted media. Transcription levels were normalized using *ACT1* gene as an endogenous reference. Group I: non-mitochondria-related candidates. Group II: mitochondria-related candidates. Error bars, mean ±SEM. (C) Western blot analysis of Tpi1p, Rlm1p-9myc, Atp11p and Mprl38p-9myc at indicated time points after glucose deprivation. The 9myc tag was inserted at the end of the coding sequence without affecting the 3'UTR. Pgk1p was used as a loading control. Anti-Tpi1p, anti-Atp11p, anti-myc and anti-Pgk1p were used for detection. Results are representative of 3–4 independent experiments per target protein.

DOI: https://doi.org/10.7554/eLife.29815.011

The following source data and figure supplements are available for figure 3:

**Source data 1.** Data used for plotting.

DOI: https://doi.org/10.7554/eLife.29815.013

*Figure 3 continued on next page*

*Figure 3 continued*

**Figure supplement 1.** Changes in total candidate mRNA levels and validation of pulse-chase protocol.
DOI: https://doi.org/10.7554/eLife.29815.012
**Figure Supplement 1—source data 1.** Data used for plotting.
DOI: https://doi.org/10.7554/eLife.29815.014
**Figure Supplement 1—source data 2.** Data used for plotting.
DOI: https://doi.org/10.7554/eLife.29815.015
**Figure Supplement 1—source data 3.** Data used for plotting.
DOI: https://doi.org/10.7554/eLife.29815.016
**Figure Supplement 1—source data 4.** Data used for plotting.
DOI: https://doi.org/10.7554/eLife.29815.017

Our data provide strong evidence that the decay kinetics and stability of mRNAs within P-bodies depend on individual properties, and that mRNAs acting in the same process might be co-regulated.

Next, we asked whether the fate of an mRNA has an impact on its translation product. Therefore, we assessed the protein level of Tpi1p and Rlm1 (Group I) as well as Atp11p and Mrpl38p (Group II) upon glucose depletion over time (*Figure 3C*). Consistent with the changes in mRNA levels, Group I protein levels dropped, while the Group II protein levels remained stable or increased over the glucose starvation time course. Our results reveal distinct and separable roles of P-bodies in regulating mRNA stabilities. On one hand, P-bodies contain transcripts undergoing decay in an individually regulated time-dependent manner. On the other hand, certain mRNAs, whose protein product contributes to stress response, are protected by P-bodies. It is possible, however, that other regulatory circuits operate independent of P-bodies, in particular since mRNAs could be cycling in and out of P-bodies in a relevant and dynamic manner influencing the fate of a particular mRNA.

## Puf5p contributes to both recruitment and decay of P-body mRNAs

Next, we aimed to record the transport of mRNAs into P-bodies by live-cell imaging using the well-established MS2 and U1A systems (*Chung and Takizawa, 2011*; *Zenklusen et al., 2007*). Tagging transcripts with U1A stem loops massively induced P-body formation under non-stress conditions (data not shown). Similarly, appending candidate transcripts with MS2 loops increased the co-localization of mRNA and P-body components to almost 100% (*Figure 4—figure supplement 1*), which is in marked contrast to the FISH-IF data (*Figure 2*). This high degree of co-localization can be explained by the recent finding that highly repetitive stem-loops can lead to non-degradable 3' mRNA fragments causing mislocalization of tagged mRNAs (*Garcia and Parker, 2015*). Considering the strong discrepancy between the FISH and MS2 localization data in terms of extent of P-body localization, and the recently published potential aberrant localization of MS2-tagged mRNAs, we decided to use the more conservative and less error-prone FISH-IF method to identify factors required for the localization and/or fate of mRNAs in P-bodies. We explored several known protein factors, which may contribute to this process with a candidate approach using *BSC1* (Group I) and *ATP11* (Group II) probes (*Figure 4—figure supplement 2A*). We deleted known P-body components or factors associating with P-bodies upon glucose deprivation (Sbp1p, Khd1p, Ngr1p and Whi3p) (*Cai and Futcher, 2013*; *Mitchell et al., 2013*) and candidates known to promote mRNA decay or repress mRNA translation, including poly(A)-binding protein II (Pbp2p), two PUF family proteins (Puf3p and Puf5p) and one non-canonical PUF protein (Puf6p) (*Chritton and Wickens, 2010*; *Wickens et al., 2002*). Remarkably, the loss of Puf5p efficiently inhibited the recruitment of *ATP11* to P-bodies as the co-localization dropped to background levels (*Figure 4A and B*, *Figure 4—source data 1*). In contrast, *BSC1* localization was unaffected (*Figure 4A and B*, *Figure 4—source data 1*). The observed lack of *ATP11* P-body localization in Δ*puf5* cells was specific, since none of the other deletion strains showed a targeting defect (*Figure 4—figure supplement 2A*). To investigate the consequence of the inability of *ATP11* to be protected in P-bodies in Δ*puf5*, we determined the *ATP11* mRNA levels. Indeed, *ATP11* mRNA levels declined, when no longer associated with P-bodies (*Figure 4C*). These data confirm that *ATP11* mRNA is protected in P-bodies from decay.

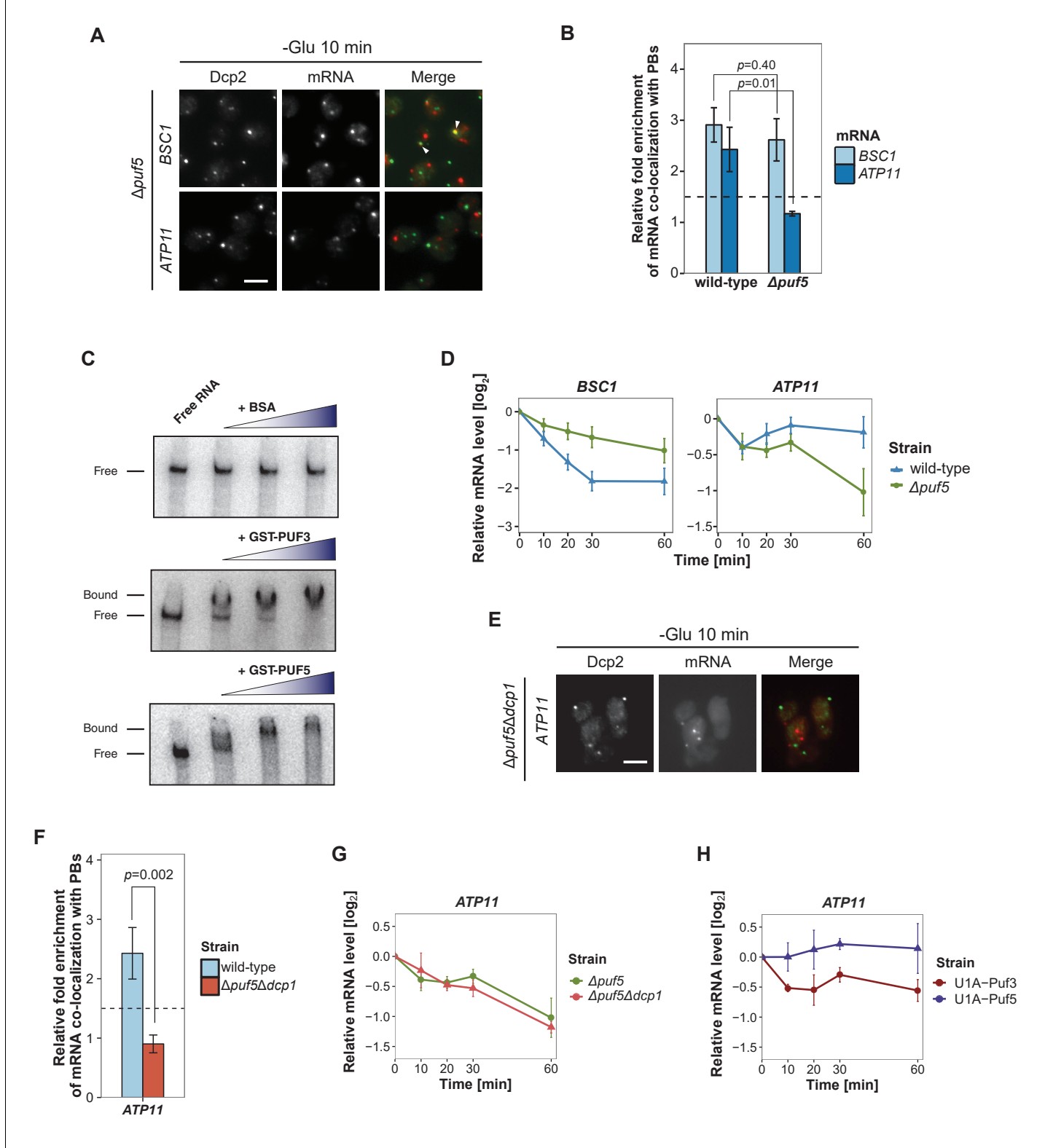

**Figure 4.** Puf5p is required for mRNA targeting to P-bodies. (**A**) Fluorescence images of P-bodies and *BSC1* (Group I) or *ATP11* (Group II) mRNAs following glucose depletion on Δ*puf5* cells expressing Dcp2-GFP. Scale bar, 5 μm. (**B**) Bar plot showing the relative fold enrichment of co-localization between *BSC1*, *ATP11* and P-bodies in Δ*puf5* strain 10 min after switched to glucose-free media. Wild type is plotted as in *Figure 2C*. The dashed line represents a fixed threshold of 1.5 for determining significant enrichment. Error bars, mean ±SEM. A one-tailed, non-paired Student's *t*-test was used to

*Figure 4 continued on next page*

*Figure 4 continued*

determine *p* values. (**C**) EMSA assays using *ATP11* 3'UTR RNA (1–500 nt after STOP codon) oligonucleotide in the absence or presence of bovine serum albumin (1.25, 2.5, 5 µM), GST-Puf3 (10, 50, 100 nM) and GST-Puf5 (1.25, 2.5, 5 µM). Unbound radiolabelled RNA (Free) shifts to a high molecular weight complex when bound to GST-Puf3 or GST-Puf5 (Bound), Results are representative of 3–4 independent experiments per protein. (**D**) The stability of 4TU labeled *BSC1* and *ATP11* mRNAs was measured by qRT-PCR in Δ*puf5* strain at indicated time points following glucose depletion. Wild type is plotted as in *Figure 3B*. Error bars, mean ±SEM. (**E**) Fluorescence images of P-bodies and *ATP11* mRNA following glucose depletion on Δ*puf5*Δ*dcp1* cells expressing Dcp2-GFP. Scale bar, 5 µm. (**F**) Bar plot showing the relative fold enrichment of co-localization between *ATP11* and P-bodies in Δ*puf5*Δ*dcp1* strain upon 10 min glucose starvation. Wild type is plotted as in *Figure 2C*. The dashed line represents a fixed threshold of 1.5 for determining significant enrichment. Error bars, mean ±SEM. A one-tailed, non-paired Student's *t*-test was used to determine *p* values. (**G**) The stability of 4TU labeled *ATP11* mRNA was measured by qRT-PCR in Δ*puf5*Δ*dcp1* strain at indicated time points following glucose depletion. Δ*puf5* is plotted as in *Figure 4D*. Error bars, mean ±SEM. (**H**) The stability of *ATP11* mRNA was examined by qRT-PCR after blocking transcription by 1, 10-phenanthroline in Dcp2-2xmcherry strains co-expressing PGK1-U1A (stem loops)-STL1 and U1A (coat protein)-GFP-Puf5 or U1A (coat protein)-GFP-Puf3.

DOI: https://doi.org/10.7554/eLife.29815.018

The following source data and figure supplements are available for figure 4:

**Source data 1.** Data used for plotting.
DOI: https://doi.org/10.7554/eLife.29815.021
**Source data 2.** Data used for plotting.
DOI: https://doi.org/10.7554/eLife.29815.022
**Source data 3.** Data used for plotting.
DOI: https://doi.org/10.7554/eLife.29815.023
**Source data 4.** Data used for plotting.
DOI: https://doi.org/10.7554/eLife.29815.024
**Source data 5.** Data used for plotting.
DOI: https://doi.org/10.7554/eLife.29815.025
**Figure supplement 1.** Live-cell detection of P-bodies (Dcp2-2xmcherry) and *BSC1* mRNA molecules using the MS2 system.
DOI: https://doi.org/10.7554/eLife.29815.019
**Figure supplement 2.** A screen for RNA-binding proteins required for mRNA recruitment to P-bodies.
DOI: https://doi.org/10.7554/eLife.29815.020
**Figure Supplement 2—source data 1.** Data used for plotting.
DOI: https://doi.org/10.7554/eLife.29815.026

Conversely, the localization of *BSC1* mRNA to P-bodies was not altered in cells lacking Puf5p, and the mRNA seemed to be stabilized to a certain degree, consistent with Puf5p's role in mRNA decay (*Goldstrohm et al., 2006*). Recent data suggest that Puf5p binds to both *BSC1* and *TPI1* mRNA, but not to any of the candidates of Group II (*Wilinski et al., 2015*). In contrast, *ATP11* has been reported to be a target of Puf3p (*Gerber et al., 2004*). However, in Δ*puf3* neither the localization to P-bodies nor *ATP11* stability was affected, suggesting Puf3p is presumably not essential for P-body related *ATP11* regulation upon glucose deprivation (*Figure 4—figure supplement 2B*, *Figure 4—figure supplements 2—source data 1*). Even though, others and we were unable to detect Puf5p in P-bodies (*Figure 4—figure supplement 2C*) (*Goldstrohm et al., 2006*), it is still possible that Puf5p interacts with *ATP11*. To address this possibility, we performed electro mobility shift assays (EMSAs) with 500 bp of the *ATP11* 3'UTR and Puf3p and Puf5p. (*Figure 4D*, *Figure 4—source data 2*). Both Puf3p and Puf5p, but not BSA bound the *ATP11* 3'UTR, albeit the Puf5p binding affinity being much weaker. Neither *ATP11* nor any of the other Group II mRNAs tested, contains a recognizable Puf5-binding sequence, indicating the presence of a non-canonical binding site. Our data so far suggest that Puf5p directly controls *BSC1* and *ATP11* mRNA stability and *ATP11* mRNA localization.

Puf5 might be required for either transport of *ATP11* to P-bodies or to protect *ATP11* from decay, or both processes. To distinguish between these possibilities, we deleted the enhancer of decapping activity, *DCP1* in a strain lacking *PUF5* (Δ*puf5* Δ*dcp1*). If Puf5p was protecting *ATP11*, then slowing down decay should restore P-body localization of *ATP11* in Δ*puf5*. However, *ATP11* mRNA levels in P-bodies were not restored (*Figure 4E and F*, *Figure 4—source data 3*). Accordingly, the decay rate of *ATP11* was essentially indistinguishable between Δ*puf5* and Δ*puf5* Δ*dcp1* (*Figure 4G*, *Figure 4—source data 4*), arguing against a protective role of Puf5p in P-bodies. Finally, we aimed to artificially localize Puf5p to P-bodies by fusing Puf5p to U1A binding protein tagged with GFP and *PGK1-U1A-STL1* RNA. Already under non-inducing conditions, some P-bodies

were induced and positive for GFP (*Figure 4—figure supplement 2D*), a phenotype, which was enhanced under glucose starvation. Under starvation about 50–60% of the U1ACP-GFP-Puf5 co-localized with Dcp2-2xmCherry. However, tethering of Puf5p to P-bodies did not affect *ATP11* mRNA stability (*Figure 4H*, *Figure 4—source data 5*). Conversely, the fusion of Puf3p to U1ACP-GFP, which was more efficiently targeted to P-bodies (*Figure 4—figure supplement 2D*), caused decay of *ATP11* mRNA (*Figure 4H-Figure 4—source data 5*) consistent with previous findings (*Miller et al., 2014*; *Olivas and Parker, 2000*). Therefore, Puf5p is likely involved in targeting, rather than locally protecting, *ATP11* mRNA to P-bodies.

## The 3'UTR is necessary but not sufficient for mRNA targeting to P-bodies

Considering that the 3'UTR of mRNAs contains most regulatory elements, which often have an important role in determining mRNA localization (*Andreassi and Riccio, 2009*; *Vuppalanchi et al., 2010*), we next investigated whether 3'UTRs play a role in mRNA targeting to P-bodies. We replaced the endogenous 3'UTR of *BSC1* and *ATP11* with the 3'UTR of *K. lactis TRP1* (*klTRP1*) and examined the localization of the chimera by FISH-IF after glucose starvation (*Figure 5A*). Replacing the 3'UTR abolished recruitment of both mRNAs to P-bodies (*Figure 5B and C*, *Figure 5—source data 1*), suggesting that even though the localization signal must be different between *ATP11* and *BSC1*, the necessary sequences are present in the 3'UTR. Consistent with the mislocalization, *BSC1* and *ATP11* transcripts were stabilized and degraded, respectively (*Figure 5D*, *Figure 5—source data 2*). The destabilization of the *ATP11* mRNA is also reflected in the reduction of Atp11p protein levels under the same conditions. Thus, the 3'UTR is essential for the fate and P-body localization under glucose starvation for both transcripts.

Since the 3'UTR was essential for both mRNAs, we investigated whether common primary sequence motifs between all mRNAs, which were specifically associated with P-body components under a unique stress, exist using the MEME Suite (*Bailey et al., 2009*). Perhaps not surprisingly, we did not find any significant primary sequence conservations or enrichment, of any particular motif. Next, we clustered stress-dependent P-body mRNAs based on secondary structures within the 3'UTR using NoFold (*Middleton and Kim, 2014*). In comparison to non-candidate mRNAs, each stress-specific candidate set contained 10–20 clusters of transcripts that were differentially enriched in certain structure motifs (*Supplementary file 2*). Interestingly, enriched motifs exhibited strong similarities (Z-score >3) to known microRNA (miRNA) motifs from RFAM, in line with the observation that at least in mammalian cells and *Drosophila*, P-bodies were shown to contain miRNA silencing complex components (*Liu et al., 2005*; *Sen and Blau, 2005*). One possible explanation is that general stem-loop structures may favor P-body localization under stress. To test this hypothesis, we determined the predicted number of stem loops in the 3'UTR of mRNAs enriched specifically under stress versus inert mRNAs and calculated the distance between stem loops. We observed a decrease in the distance between stem loops, suggesting clustering of the loops (*Figure 5—figure supplement 1A*). To determine whether clusters of stem loops would be sufficient to drive P-body localization, we transplanted the 3'UTR of *BSC1* or *ATP11* to a non-P-body associated transcript *SEC59* and a sodium specific P-body component-associated transcript *YLR042C* (*Figure 5—figure supplement 1B*). None of the four chimaeras recapitulated the localization of native *BSC1* and *ATP11* transcripts under stress (*Figure 5—figure supplement 1C and D*, *Figure 5—figure supplement 1—source data 1*). Thus, although the 3'UTRs are essential, they are not sufficient by themselves to drive mRNA transport into P-bodies. Most likely other elements in the coding sequence and/or 5'UTR act cooperatively.

## Overexpression of *ATP11* rescues the glycogen accumulation deficiency in Δ*puf5* cells

Finally, we asked whether the stabilization of *ATP11* mRNA by Puf5p is beneficial for the cell. Puf5p promotes chronological lifespan (*Stewart et al., 2007*), which is dependent on the accumulation of carbohydrates such as glycogen (*Cao et al., 2016*). Similarly, a Δ*atp11* strain reportedly showed decreased glycogen accumulation (*Wilson et al., 2002*). Therefore, we asked whether Atp11p levels would contribute to the Puf5p ability to promote lifespan and stained for glycogen when cells reached stationary phase. As expected, Δ*atp11* and Δ*puf5* failed to efficiently accumulate glycogen

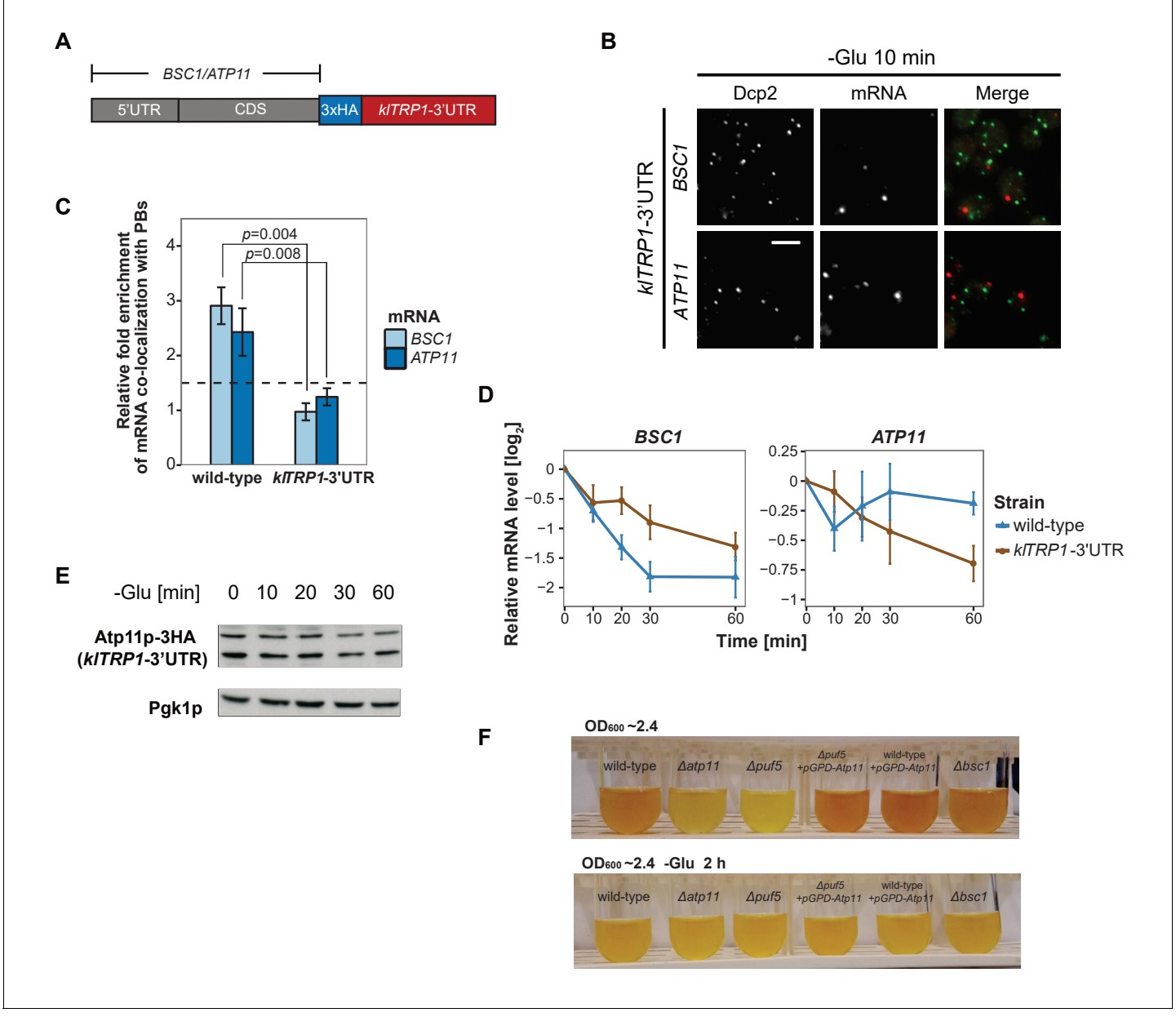

**Figure 5.** 3'UTR is necessary for mRNA localization to P-bodies. (A) A schematic representation of C-terminal tagging with 3xHA. The endogenous 3'UTR was simultaneously replaced by the 3'UTR of klTRP1. (B) Fluorescence images of P-bodies and *BSC1*, *ATP11* mRNAs following glucose depletion on corresponding 3'UTR replaced strains. Scale bar, 5 μm. (C) Bar plot depicting the relative fold enrichment of co-localization between *BSC1*, *ATP11* and P-bodies in corresponding 3'UTR replaced strains 10 min after glucose starvation. Wild type is plotted as in *Figure 2C*. The dashed line represents a fixed threshold of 1.5 for determining significant enrichment. Error bars, mean ±SEM. A one-tailed, non-paired Student's *t*-test was used to determine *p* values. (D) The stability of 4TU labeled *BSC1* and *ATP11* mRNAs was determined by qRT-PCR in corresponding 3'UTR replaced strains at indicated time points following glucose depletion. Wild type is plotted as in *Figure 3B*. Error bars, mean ±SEM. (E) Western blot analysis of Atp11p-HA (*klTRP1* 3'UTR) at indicated time points after glucose deprivation. Pgk1 was used as a loading control. Anti-HA and anti-Pgk1p were used for detection. Results are representative of three independent experiments. (F) Assessment of intracellular glycogen content in wild type, *ATP11*, *PUF5* deletion strains in the absence or presence of *ATP11* overexpression plasmid and *BSC1* deletion strain by iodine staining. Yeast cultures were grown to stationary phase (OD$_{600}$ ~2.4) in medium containing 2% dextrose (upper panel). Then cells were shifted to medium without dextrose for 2 hr (lower panel). Results are representative of four independent experiments.

DOI: https://doi.org/10.7554/eLife.29815.027

The following source data and figure supplements are available for figure 5:

**Source data 1.** Data used for plotting.
DOI: https://doi.org/10.7554/eLife.29815.029

*Figure 5 continued on next page*

*Figure 5 continued*

**Source data 2.** Data used for plotting.
DOI: https://doi.org/10.7554/eLife.29815.030
**Figure supplement 1.** 3'UTR is insufficient for mRNA localization to P-bodies.
DOI: https://doi.org/10.7554/eLife.29815.028
**Figure Supplement 1—source data 1.** Data used for plotting.
DOI: https://doi.org/10.7554/eLife.29815.031

as indicated by the absence of the brown color (*Figure 5F*). Importantly overexpression of *ATP11* in the *Δpuf5* strain was sufficient to restore glycogen accumulation, suggesting that the stabilization of *ATP11* mRNA by Puf5p contributes to Puf5p's positive effect on chronological lifespan.

## Discussion

The fate of mRNAs and their regulation under different stress conditions is still not well understood. mRNAs have been proposed to be either associated with ribosomes or stored/decayed in P-bodies and SGs. Here we demonstrate that the content and the fate of mRNA in P-bodies is stress-dependent, varying from decay to stabilization. We furthermore provide evidence that different mRNA classes use different mechanisms to be P-body localized. The localization and fate of these mRNAs are dependent on interactions with RNA binding proteins such as Puf5p and essential information present in the 3'UTR of the mRNA.

To enable this analysis, we first devised a method to enrich RNPs based on in vivo chemical cross-linking followed by streptavidin affinity purification. This method allows the identification and global analysis of mRNAs associated with P-body components. We previously used a similar approach to successfully discover a novel exomer-dependent cargo (*Ritz et al., 2014*), novel interactors of the ArfGAP Glo3 (*Estrada et al., 2015*) and a novel facultative P-body component (*Weidner et al., 2014*). We improved the procedure permitting the reliable enrichment and detection of mRNAs associated with membrane-localized P-body components under a variety of stress conditions. Furthermore, our method works regardless of poly(A) tail length or partial transcript degradation, and hence could be applied for the identification of many types of RNAs. Moreover, this method would also be applicable to study protein-DNA interactions.

We mostly concentrated our subsequent analysis on hits from the glucose starvation experiments but it is very likely that these findings can be generalized to other stresses. We identified three classes of mRNAs associated with P-body components at membranes. The first class consists of mRNAs that are generally deposited into P-bodies, independent of the stressor. We did not investigate their fate further in this study, but we assume that most of those transcripts would be prone to decay. The second class contains mRNAs that are stressor-dependent and decayed. It is important to note that the decay rate of mRNAs in this class is very variable and could represent an intrinsic property of the mRNA or a subset of mRNAs. Some transcripts will be decayed almost immediately after arrival in P-bodies, while others are initially excluded from degradation. The kinetics of decay also appears to vary, indicating that even within P-bodies the degradation of client RNAs is highly regulated. Finally, the third class corresponds to mRNAs that are also stress-specific, but are stabilized, rather than being degraded. It appears as if this class is enriched in transcripts whose products would be beneficial for stress survival. This hypothesis is based on the stabilization of transcripts involved in mitochondrial function under glucose starvation, a condition under which mitochondria are up-regulated (*Wu et al., 2004*). A recent study also suggested stabilization of transcripts in a P-body-dependent manner under high-osmolarity stress (*Huch and Nissan, 2017*). Thus, P-bodies emerge as context-dependent regulator in stress responses. Although P-bodies have been proposed previously as sites of mRNA decay and storage (*Sheth and Parker, 2003*), the studies on which this model was based had either been performed on very few selected transcripts or artificial transcripts with extended G-tracts driving P-body localization through imaging or genome-wide analyses, taking all the mRNAs present in the lysate into account (*Arribere et al., 2011*; *Brengues et al., 2005*; *Sun et al., 2013*). Our approach is different in that we enrich first for P-bodies and then extract the RNA specifically from the P-body fraction. Therefore, our data provide an unprecedented wealth of information on the mRNA content and fate within P-bodies.

Since the fate of an mRNA is stressor-dependent, it is tempting to speculate that the different mRNA classes are recruited to P-bodies through different pathways. In support of this hypothesis, we identified the RNA binding protein Puf5p as a protein regulating both the localization of one transcript, as well as the degradation of another (*Figure 6*). The latter function is easily explained by the established role of Puf5p as interactor of the Ccr4/Not deadenylation complex, which shortens the poly(A)-tail independent of the subsequent route of destruction through P-bodies or exosomes (*Balagopal et al., 2012*). In fact, *BSC1* mRNA was recently identified as Puf5p target (*Wilinski et al., 2015*). In the absence of Puf5p, *ATP11* is no longer P-body localized and is destabilized. Hence, in this case, P-bodies protect an mRNA from degradation in a Puf5p-dependent manner. It is tempting to speculate that the P-body localized *ATP11* mRNA is protected from interaction with Puf3p, which would be able to trigger *ATP11* destruction (*Miller et al., 2014*). It is striking, however, that Puf5p possesses this dual role in stabilization and destruction depending on the client mRNA, as well as being involved in the localization of mRNAs to P-bodies.

The notion that mRNAs are decayed in P-bodies was recently challenged (*Pelechano et al., 2015*; *Sweet et al., 2012*). Instead, it was suggested that decay might mostly happen co-translationally. We cannot exclude that a part of the RNAs is degraded co-translationally, since the decay machinery in both processes appears to be identical. In favor of mRNA decay in P-bodies, we confirmed hits from the biochemical enrichment procedure by in vivo localization studies. We found that

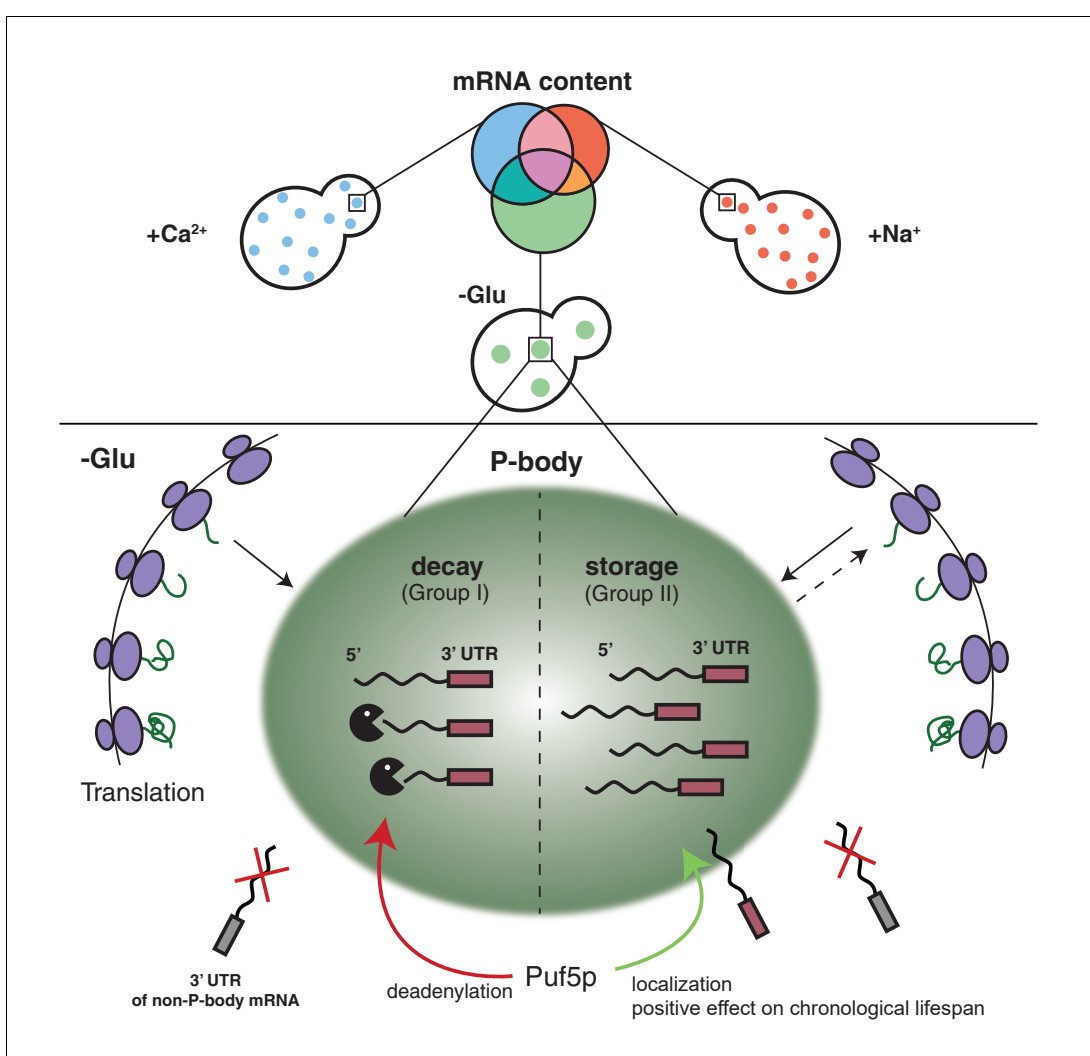

**Figure 6.** Schematic model summarizing our findings.
DOI: https://doi.org/10.7554/eLife.29815.032

P-body-localized mRNAs were degraded with different kinetics. Moreover, we would expect to find significantly higher sequence coverage of the 3' region of candidate mRNAs, which we did not observe. Also, the fate -stabilization versus degradation- of *BSC1* in a Puf5p-dependent manner, which was not accompanied by modulating P-body localization, is in support of P-body as decay compartment. Thus, our data are consistent with mRNA degradation in P-bodies under stress conditions. In contrast, *ATP11* may become a co-translational degradation target in the absence of Puf5p. However co-translational mRNA decay might still be a major pathway in non-stressed cells, in which microscopically visible P-bodies are not frequently detected. At least the 5' decay machinery, the helicase Dhh1p and the 5'exonuclease Xrn1p have been found to be associated with polysomes also in the absence of a stressor (*Pelechano et al., 2015*; *Sweet et al., 2012*; *Weidner et al., 2014*).

Our findings demonstrate that P-body associated mRNA can follow different fates, namely decay or stabilization. Whether these two functions are performed by the same or different P-bodies remains unclear. We favor the possibility, however, that both functions can be provided by the same P-body. Recent data from *Drosophila* sponge bodies, which are the equivalent of P-bodies in embryos, suggest that storage and decay may happen in the same compartment (*Weil et al., 2012*). Likewise, there is no evidence thus far for differential protein composition of P-bodies formed under the same stress condition (*Kulkarni et al., 2010*). Although, it is possible that the transient protein components may vary from one another, we expect the major factors would be discriminative to fulfill opposing functions and mRNA selectivity.

Stabilized mRNAs may return into the translation competent pool. So far, we cannot exclude that the stabilized mRNA is cycling in and out of P-bodies dynamically and whether this is part of the stabilization process. The finding that P-body localized Puf3p triggered *ATP11* destruction would argue against dynamic cycling being a requirement for stabilization; future experiments are needed to clarify this point. Furthermore, whether this re-initiation would be through diffusion of the mRNA from the P-body into the cytoplasm or through another organelle, such as stress granules (SGs), remains to be established. SGs harbor stalled translation initiation complexes, whose formation can also be triggered upon a variety of stresses. Additionally, SGs frequently dock and fuse with P-bodies, and they share some common protein factors (*Buchan et al., 2008*; *Buchan et al., 2011*; *Kedersha et al., 2005*; *Stoecklin and Kedersha, 2013*). As mRNAs in SGs are polyadenylated, they are not subject to immediate degradation (*Kedersha et al., 1999*; *Stoecklin and Kedersha, 2013*). Based on those evidences, we speculate that the re-engagement of stable transcripts into translation is likely mediated via SGs.

A number of genome-wide studies detailing responses to stress have been performed (*Miller et al., 2011*; *Munchel et al., 2011*). Most of the studies deal with global RNA synthesis and decay, but do not provide any insights into the regulated storage of mRNA. In this study, we addressed this issue and uncovered Puf5 as key molecule in the decision-making whether or not a particular mRNA must be degraded under glucose starvation. This decision-making explains Puf5p's positive effect on chronological lifespan, as increasing Atp11p levels were sufficient to rescue the glycogen accumulation defect of Δ*puf5* cells. How the decision making is brought about will be the focus of future studies.

## Materials and methods

**Key resources table**

| Reagent type (species) or resource | Designation | Source or reference | Identifiers | Additional information |
|---|---|---|---|---|
| strains are listed in *Supplementary file 5* | | | | |
| primers are listed in *Supplementary file 4* | | | | |
| genetic reagent (Plasmid) | pDZ274 | Addgene plasmid # 45929 | | |
| genetic reagent (Plasmid) | pDZ415 | Addgene plasmid # 45162 | | |
| antibody | anti-DIG-POD | Roche | RRID:AB_514500 | 1/750 in PBTB |
| antibody | anti-HA | Covance | RRID:AB_2314672 | 1/250 FISH-IF, 1/1,000 WB |
| antibody | anti-GFP | Roche | RRID:AB_390913 | 1/250 |

*Continued on next page*

*Continued*

| Reagent type (species) or resource | Designation | Source or reference | Identifiers | Additional information |
|---|---|---|---|---|
| antibody | goat anti-mouse-IgG-Alexa 488 | Invitrogen | RRID:AB_2534069 | 1/400 in PBS |
| antibody | anti-Tpi1 | LSBio | RRID:AB_11132833 | 1/1,000 |
| antibody | anti-myc | Sigma-Aldrich | RRID:AB_439694 | 1/1,000 |
| antibody | anti-Pgk1 | Invitrogen | RRID:AB_221541 | 1/1,000 |

## Yeast strains and growth conditions

Standard genetic techniques were employed throughout (*Sherman, 1991*). Unless otherwise noted, all genetic modifications were carried out chromosomally. Chromosomal tagging and deletions were performed as described (*Janke et al., 2004*; *Knop et al., 1999*). For C-terminal tagging with 3xHA, the plasmid pYM-3HA (*klTRP1*) and with 9xmyc the plasmid pOM20 (*kanMX6*) and pSH47 (*URA3*) were used. The use of pOM plasmids (*Gauss et al., 2005*) in combination with Cre recombinase allowed C-terminal chromosomal tagging and preservation of the endogenous 3'UTR at the same time. The plasmid pFA6a-natNT2 was used for construction of all deletion strains, except for Δ*puf3* (pUG73), Δ*dcp1* (pUG73), Δ*atp11* (pUG72) and Δ*bsc1* (pUG72). 3'UTR transplantation experiments were carried out with the *Delitto Perfetto* method using the pCORE plasmid (*kanMX4-URA3*) (*Storici and Resnick, 2006*).

For C-terminal tagging of Puf5p, Tif4632p and Pub1p with GFP, the plasmid pYM26 (*klTRP1*) was used. pFA6a-3xmcherry (*hphNT1*) plasmid was used in tagging Dcp2p with mcherry (*Maeder et al., 2007*). For MS2 live-cell mRNA imaging, MS2SL tagged strains were constructed using pDZ415 (24MS2SL loxP-Kan-loxP). To remove selection marker and visualize the transcripts, the Cre recombinase-containing plasmid pSH47 (URA3) and MS2SL coat protein expressing plasmid pDZ274 (pLEU MET25pro MCP-2x-yeGFP) were co-transformed into cells (*Hocine et al., 2013*). Dcp2-2xmcherry strains were co-transformed with plasmids expressing PGK1-U1A (stem loops)-STL1 and U1A (coat protein)-GFP-Puf3 or U1A (coat protein)-GFP-Puf5 for live-cell imaging with the U1A system. PGK1-U1A (stem loops)-STL1 was created by replacing *PGK1* 3'UTR of pPS2037 with *STL1* 3'UTR (1–500 nt after STOP codon). U1A-GFP-Puf3 and U1A-GFP-Puf5 plasmids were constructed by inserting the *PUF3* or *PUF5* ORF into pRP1187. A short linker (Gly)eight was introduced between GFP and the Puf3 or Puf5. Plasmids pDZ415 (Addgene plasmid # 45162) and pDZ274 (Addgene plasmid # 45929) were gifts from Robert Singer and Daniel Zenklusen (Albert Einstein College of Medicine, Bronx, NY, USA). Primers and strains used in this study are listed in *Supplementary file 4* and *Supplementary file 5*.

Unless otherwise noted, yeast cells were grown in YPD (1% yeast extract, 2% peptone, 2% dextrose) at 30 ˚C. For glucose deprivation, cultures were further grown in YP media without dextrose for indicated times. For mild osmotic stress, YPD growth medium was supplemented with 0.5 M NaCl or 0.2 M $CaCl_2$ for indicated times. Yeast cells were harvested at mid-log phase ($OD_{600}$ of 0.4–0.8).

## Chemical cross-linking coupled to affinity Purification (cCLAP) and preparation of RNA-Seq samples

The cCLAP was carried out according to *Tagwerker et al., 2006*, *Hafner et al. (2010)* and *Kishore et al. (2011)* with modifications. Cells expressing Dcp2-HBH or Scd6-HBH were grown to mid-log phase, subjected to the corresponding stress and crosslinked with 1% formaldehyde for 2 min and quenched with 125 mM glycine for 10 min. Control cells were treated equally except stress application. Cells were lysed in RIPA buffer (50 mM Tris-HCl pH 8.0, 150 mM NaCl, 1% NP-40, 0.5% sodium deoxycholate, 0.1% SDS, supplemented with protease inhibitors) using a FastPrep (MP Biomedicals, Santa Ana, CA). Cell debris was removed by a low speed spin (1300 x g, 5 min 4°C). To dissolve large RNPs, supernatants were treated with 50 U/ml RNase T1 (Fermentas, Waltham, MA) at 22°C for 15 min. Lysates were spun at 20,000 x g for 10 min and 4°C, and the pellets resuspended in binding buffer (50 mM NaPi pH 8.0, 300 mM NaCl, 6 M GuHCl, 0.5% Tween-20). Pull-downs were performed with streptavidin agarose beads (Thermo Fisher Scientific, Waltham, MA). The second RNase T1 digestion was performed on the beads with a final concentration of 1 U/μl. Radiolabeling of RNA was performed by adding 0.5 μCi/μl γ-$^{32}$P-ATP (Hartmann analytic, Germany) and 1 U/μl T4

PNK (New England Biolabs, Ipswich, MA). To purify RNA, proteins were digested using 1.2 mg/ml proteinase K (Roche, Switzerland) in 2 x proteinase K buffer (100 mM Tris-HCl pH 7.5, 200 mM NaCl, 2 mM EDTA, 1% SDS) for 30 min at 55 ˚C. The RNA was subsequently isolated using phenol-chloroform-isoamyl alcohol (125:24:1) (Sigma-Aldrich, Germany) as described (*Schmitt et al., 1990*) followed by 2 hr incubation at 65 ˚C to reverse formaldehyde crosslinking. Purified RNA was subjected to 3' and 5' adapter ligation following Illumina's TruSeq Small RNA Library Prep Guide. To reduce the rRNA species, RiboMinus transcriptome isolation kit (Invitrogen, Waltham, MA) was used according to the manufacturer's protocol. Reverse transcription using SuperScript III reverse transcriptase (Invitrogen), oligo-dT and random hexamer was performed afterwards. The cDNA libraries were generated by a final PCR amplification step with llumina indexing primer (RPI1-4, *Supplementary file 4*).

In this study, five library sets (from five biological replicates) were sequenced. Except the first library set, all the libraries were generated as described above. In the first library set, the radiolabeling step was omitted and the PAGE purification steps were replaced by column-based purification with RNeasy kit (Qiagen, Germany), according to the manufacturer's instruction.

## Preparation of total RNA-Seq libraries

Cells expressing Dcp2-HBH were grown to mid-log phase, and lysed followed by total RNA isolation using phenol-chloroform-isoamyl alcohol (125:24:1) (Sigma-Aldrich) as described (*Schmitt et al., 1990*). Library preparation was performed with 1 µg total RNA using the TruSeq Stranded mRNA Library Prep Kit with D501-D508, D701 and D702 adapters (Illumina, San Diego, CA). Three libraries (from three biological replicates) were sequenced.

## Processing of small RNA-Seq reads

RNA-Seq libraries were sequenced on Illumina HiSeq2000 with single read to 50 bp. We clipped adapters and trimmed low quality bases using Trimmomatic version 0.30 (RRID:SCR_011848) (*Bolger et al., 2014*) with parameters 'SE –s phred33 ILLUMINACLIP:llumina_smallRNA_adapters. fa:1:20:5 LEADING:30 TRAILING:30 MINLEN:10', where Illumina_smallRNA_adapters.fa contained all adapter and primer sequences from the TruSeq Small RNA Sample Preparation Kit. Subsequently, reads were aligned to *Saccharomyces cerevisiae* genome EF4.72 from ENSEMBL using Bowtie version 1.0.0 (RRID:SCR_005476) (*Langmead et al., 2009*) with parameters '-n 0 –l 28 –e 70 –k1 –m 1 –best –strata –sam –nomaqround'. Reads were counted per exon using htseq-count (RRID:SCR_011867) (*Anders et al., 2015*) with default parameters against ENSEMBL's matching GTF file for EF4.72 and aggregated on the gene-level. The workflow is summarized in *Figure 1—figure supplement 2*.

## Analysis of P-body associated mRNAs and total RNA-Seq

Analysis of P-body associated mRNAs was performed using edgeR versions 3.0 and 3.12.1 (RRID:SCR_012802) (*Robinson et al., 2010*) using standard procedures for count normalization and estimation of dispersion. The gel label and batch were included as factors in the experimental design (*Supplementary file 1*). We identified significantly ($p < 0.05$) upregulated mRNAs exclusive for each stress condition by testing each individual stress condition against the wild type condition and removing those mRNAs that were identified as common hits when testing the joint set of stress conditions against unstressed control. For glucose depletion, we additionally excluded genes previously shown to be significantly enriched in polysomes (*Arribere et al., 2011*) for the same stress.

Total RNA-Seq libraries were sequenced on Illumina NextSeq500 with single read to 76 bp. The same processing and analysis pipeline was applied as above. Only protein coding RNA species were taken into account. Experimental design is shown in *Supplemental File 3*.

## Gene Ontology (GO) term enrichment analysis

mRNAs associated with P-body components for each stress condition were tested for GO biological processes (BP) enrichment using hypergeometric tests as implemented in the hyperGTest function from the GOstats R/Bioconductor package version 1.7.4. (RRID:SCR_008535). The mRNA universe was defined for each stress condition as the set of mRNAs with a mean expression over all replicates larger than or equal to the first quartile. For GO term mRNA annotation, the R/Bioconductor package org.Sc.sgd.db version 3.1.2 was used. p-Values from the hypergeometric tests were visualized

using the ggplot2 R package version 1.0.1 (RRID:SCR_014601). The workflow is summarized in *Figure 1—figure supplement 2*.

## Comparison of mRNA lengths

mRNA (total exonic gene) lengths were extracted by using R/Bioconductor package GenomicFeatures version 1.22.13 (RRID:SCR_006442). A non-parametric, two-sided Wilcoxon test was used to determine *p* values.

## Combined fluorescence in situ hybridization (FISH) and immunofluorescence (IF)

Combined FISH and IF was performed as described (*Kilchert et al., 2010*; *Takizawa et al., 1997*). The following antibodies and solution were used for detection: anti-DIG-POD (RRID:AB_514500, Roche, 1:750 in PBTB), anti-HA (RRID:AB_2314672, Covance, Princeton, NJ, HA11; 1:250), anti-GFP (RRID:AB_390913, Roche GFP clones 7.1 and 13.1, 1:250), goat anti-mouse-IgG-Alexa488 (RRID:AB_2534069, Invitrogen, 1:400 in PBS) and tyramide solution (PerkinElmer, Waltham, MA, 1:100 in Amplification Solution supplied with kit). Primers with T7 promoter ends (*Supplementary file 4*) and MEGAscript T7 transcription kit (Ambion, Waltham, MA) were used for probe generation. To obtain fluorescence images, slides were mounted with Citifluor AF1 (Citifluor, Hatfield, PA), supplemented with 1 µg/ml DAPI to stain the nuclei. Images were acquired with an Axiocam MRm camera mounted on an Axioplan two fluorescence microscope using a Plan Apochromat 63x/NA1.40 objective and filters for eqFP611 and GFP. Axiovision software 3.1 to 4.8 (RRID:SCR_002677) was used to process images (Carl Zeiss, Germany).

## Co-localization analysis

Signals of P-bodies and mRNA were identified using the spots tools in Imaris software package (Bitplane) (RRID:SCR_007370). For co-localization determination, the MATLAB (MathWorks)- Imaris plug-in 'co-localize spots' function was used. Two spots were defined as co-localized spots, if the distance between two centers of the spots was smaller than the radius of the smaller spot among the two. The percentage of mRNA co-localization with P-bodies was calculated by dividing co-localized FISH spots by total FISH spots. Approximately 200 cells from at least three biologically independent experiments were counted per mRNA per condition.

## Pulse-chase labeling with 4TU and RNA purification

The pulse-chase labeling experiment was carried out as described previously (*Zeiner et al., 2008*). For the pulse, yeast culture was grown in HC-Ura drop-out media supplemented with 2% dextrose, 0.1 mM uracil and 0.2 mM 4-Thiouracil (Sigma-Aldrich) for 5-6 hr. Yeast were spun down at 3,000 g for 2 min and resuspended in HC-Ura drop-out media containing 20 mM or 32 mM uracil (chase). Afterwards, yeasts were collected by centrifugation at the following time points: t = 0, 10, 20, 30, and 60 min. Cells were lysed followed by total RNA isolation using phenol-chloroform-isoamyl alcohol (125:24:1) (Sigma-Aldrich). The RNA was then subjected to biotinylation and further purification according to *Zeiner et al., 2008*.

The same pulse-chase labeling protocol was performed to determine the mRNA stability of *ACT1*, *PGK1* and *RPL37b* under glucose deprivation condition, and 200 pg humanized Renilla luciferase (*hRLuc*) RNA spike-in was added per microgram total RNA as reference gene. The same RNA purification protocol was followed to isolate 4TU labeled RNA as well as total RNA. At least three biologically independent pulse-chase experiments per mRNA per strain were performed.

## Transcription shut-off assay

Transcription was inhibited by adding 1,10-phenanthroline (Sigma-Aldrich) to yeast cultures to a final concentration of 100 µg/ml after glucose depletion. Yeasts were collected at indicated time points for RNA isolation using phenol-chloroform-isoamyl alcohol (125:24:1) (Sigma-Aldrich). At least three biologically independent experiments were performed.

## Quantitative RT-PCR

0.5–1 µg of 4-TU labeled RNA or total RNA was reverse transcribed with the Transcriptor reverse transcriptase kit (Roche), oligo-dTs and random hexamers. The mRNA levels were analyzed by SYBR green incorporation using ABI StepOne Plus real-time PCR system (Applied Biosystems). Primers used in qRT-PCR are listed in **Supplementary file 4**.

## Western blotting

Glucose-deprived cells were harvested at indicated times. For each time point, 9 ml of culture was collected, immediately treated with cold trichloroacetic acid (10% final concentration), and incubated on ice for 5 min. Yeast extracts were prepared as described (**Stracka et al., 2014**). The protein concentration was determined using the DC Protein Assay (Bio-Rad), and the total lysate was analyzed by SDS-PAGE and immunoblotting. The following antibodies were used for immunoblotting: anti-Tpi1p (RRID:AB_11132833, LSBio LS-C147665;); anti-Atp11p (a gift from Sharon H. Ackerman, Wayne State University, Detroit, MI); anti-HA (RRID:AB_2314672, Covance HA11; 1:1,000); anti-myc (RRID:AB_439694, M4439; Sigma-Aldrich; 1:1,000); anti-Pgk1p (RRID:AB_221541, Invitrogen #A-6457; 1:1,000). Enhanced Chemiluminescence (ECL; GE Healthcare) was used for detection.

## Live-cell imaging

For live-cell imaging with MS2 system, yeast cells were grown in HC-Leu medium containing 2% glucose to mid-log phase. The cells were taken up in glucose-free HC-Leu medium afterwards. For the U1A system, HC-Ura-Trp medium was used. For live-cell imaging of GFP- or mCherry-tagged fusions of Dcp2p, Puf5p, Pub1p and Tif4632p, yeast cells were grown in YPD medium to mid-log phase, and resuspended in HC-complete medium lacking glucose. Fluorescence was monitored as described in the FISH-IF section.

## Electrophoretic mobility shift assays (EMSA)

Recombinant GST-Puf3 (amino acids 465–879) and GST-Puf5 (amino acids 126–626) expressed form pWO12 and pWO18 (Gifts from Wendy M. Olivas, University of Missouri St. Louis, St. Louis, MO), respectively were purified and stored in 50 mM Tris/HCl pH 8.0, 10% glycerol. The *ATP11* 3'UTR RNA (1–500 nt after STOP codon) was transcribed from a template containing T7 RNA polymerase promoter with MEGAscript T7 transcription kit (Ambion) and $\alpha$-$^{32}$P-UTP (10 mCi/ml). Binding reactions (20 µL) contained 4,000 cpm of labeled RNA, varying concentrations of protein, 20 U RNasin Plus RNase Inhibitor (Promega) and 1 x binding buffer (10 mM Tris/HCl pH 7.5, 100 mM KCl, 1 mM EDTA, 0.1 mM DTT, 0.01 mg/ml bovine serum albumin, 5% glycerol). Reactions were incubated at RT for 30 min, and separated on a 4% non-denaturing acrylamide gel. Gels were dried, exposed to a phosphor screen for 10–16 hr, and the screens scanned using a phosphorimager (Typhoon FLA 7000, GE Healthcare).

## Identification of secondary structure motifs within the 3'UTRs of P-body-associated mRNAs

Secondary structure motifs in the 3' untranslated regions (UTRs) of transcripts, overrepresented among differentially enriched mRNAs for each stress condition, were identified using NoFold (**Middleton and Kim, 2014**) version 1.0. 3' UTR sequences were extracted from the biomart project (RRID:SCR_002987) (http://biomart.org) by selecting 300 base pairs (bp) downstream of the coding sequence (CDS). The internal NoFold boundary file bounds_300seq.txt was used along with a file containing UTR sequences of all non-enriched mRNAs as a background for enrichment analysis and parameter –rnaz. All other parameters were used in the default setting.

## Analysis of intracellular glycogen

Glycogen content in yeast cells was visualized using iodine staining (**Quain and Tubb, 1983**). Wild type, Dcp2-GFP $\Delta atp11$, Dcp2-GFP $\Delta puf5$ and Dcp2-GFP $\Delta bsc1$ strains were grown in HC medium, and strains containing the *ATP11* overexpression plasmid were grown in HC-Ura medium. All strains were all1owed to reach stationary phase (OD$_{600}$ ~2.4) and subsequently shifted for 2 hr to medium lacking glucose. Samples were taken before and after dextrose depletion, iodine (Sigma-Aldrich)

was added to a final concentration of 0.5 mg/ml iodine. The intensities of produced yellow-brown stain positively correlate with their intracellular glycogen levels.

## Accession numbers

The RNA-Seq data reported in this study are deposited in the Gene Expression Omnibus (GEO) database (RRID:SCR_005012), and the accession number is GSE76444.

# Acknowledgements

We thank C Brown, M Zavolan and J Guimaraes for help with the data analysis and discussions, and M Zavolan, I G Macara, T Gross, R P Jansen and W Filipowicz for critical comments on the manuscript. S Ackerman and W Olivas are acknowledged for providing the Atp11 antibody and the *PUF3* and *PUF5* constructs, respectively. This work was supported through grants from HFSP (RGP0031), the Swiss National Science Foundation (31003A_141207, 310030B_163480) and the University of Basel to AS. CW and JW were supported by Werner Siemens Fellowships and FS by SystemsX.ch (2009/025), the Swiss Initiative in Systems Biology.

# Additional information

## Funding

| Funder | Grant reference number | Author |
|---|---|---|
| Schweizerischer Nationalfonds zur Förderung der Wissenschaftlichen Forschung | 31003A_141207 | Congwei Wang Julie Weidner Anne Spang |
| Universität Basel | | Congwei Wang Julie Weidner Anne Spang |
| Human Frontier Science Program | RG0031/2009 | Congwei Wang Julie Weidner Anne Spang |
| Schweizerische Initiative in Systembiologie, SystemsX.ch | 2009/025 | Fabian Schmich Niko Beerenwinkel |
| Schweizerischer Nationalfonds zur Förderung der Wissenschaftlichen Forschung | 3100308_163480 | Congwei Wang Anne Spang |
| Werner Siemens Foundation | | Congwei Wang Julie Weidner |

The funders had no role in study design, data collection and interpretation, or the decision to submit the work for publication.

## Author contributions

Congwei Wang, Conceptualization, Formal analysis, Validation, Investigation, Visualization, Methodology, Writing—original draft, Writing—review and editing; Fabian Schmich, Formal analysis, Investigation, Methodology, Writing—review and editing; Sumana Srivatsa, Data curation, Formal analysis, Writing—review and editing; Julie Weidner, Investigation, Methodology, Writing—review and editing; Niko Beerenwinkel, Supervision, Funding acquisition, Writing—review and editing; Anne Spang, Conceptualization, Formal analysis, Supervision, Funding acquisition, Writing—original draft, Project administration, Writing—review and editing

## Author ORCIDs

Congwei Wang http://orcid.org/0000-0003-2277-5002
Niko Beerenwinkel http://orcid.org/0000-0002-0573-6119
Anne Spang http://orcid.org/0000-0002-2387-6203

**Decision letter and Author response**
Decision letter https://doi.org/10.7554/eLife.29815.042
Author response https://doi.org/10.7554/eLife.29815.043

## Additional files

### Supplementary files

• Supplementary file 1. Experimental design.
DOI: https://doi.org/10.7554/eLife.29815.033

• Supplementary file 2. Hit list RNAseq from RNAs associated with PB components.
DOI: https://doi.org/10.7554/eLife.29815.034

• Supplementary file 3. Hit list RNAseq from total RNA.
DOI: https://doi.org/10.7554/eLife.29815.035

• Supplementary file 4. List of primers used in this study.
DOI: https://doi.org/10.7554/eLife.29815.036

• Supplementary file 5. List of strains used in this study.
DOI: https://doi.org/10.7554/eLife.29815.037

• Transparent reporting form
DOI: https://doi.org/10.7554/eLife.29815.038

### Major datasets

The following dataset was generated:

| Author(s) | Year | Dataset title | Dataset URL | Database, license, and accessibility information |
|---|---|---|---|---|
| Congwei Wang, Fabian Schmich, Sumana Srivatsa, Niko Beerenwinkel, Anne Spang | 2017 | RNA-seq to identify P-body associated mRNAs upon glucose starvation, Na+ and Ca2+ stresses | https://www.ncbi.nlm.nih.gov/geo/query/acc.cgi?acc=GSE76444 | Publicly available at the NCBI Gene Expression Omnibus (accession no. GSE76444) |

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
