## [Decision Letter]

Thank you for submitting your article "Context-dependent deposition and regulation of mRNAs in P-bodies" for consideration by *eLife*. Your article has been reviewed by two peer reviewers, one of whom is a member of our Board of Reviewing Editors, and the evaluation has been overseen by James Manley as the Senior Editor. The following individual involved in review of your submission has agreed to reveal his identity: Roy Parker (Reviewer #2).

The reviewers have discussed the reviews with one another and the Reviewing Editor has drafted this decision to help you prepare a revised submission.

Summary:

In this manuscript, Spang and colleagues apply chemical cross- linking in combination with affinity purification (cCLAP – chemical Cross-Linking coupled to Affinity Purification) with the goal to identify mRNAs that enrich in P-bodies in different stress conditions. The authors use cCLAP in glucose starvation, and upon CaCl2 and NaCl salt stress. They identify more than 1500 mRNAs, which can be classified into mRNAs that are found in all stress condition and mRNAs that are stress specific. A subset of the identified mRNAs is further analyzed to gain insight into how those mRNAs accumulate in P-bodies and what the consequences are of such P-body localization.

The strengths of this manuscript are that it addresses important and interesting issues as to the composition and function of RNP granules. The weakness of the work is that some of the analyses are limited in their methodology and that some results are overinterpreted. Although there is interest from the reviewers, for acceptance in *eLife*, it will be critical to address these weaknesses (detailed below) to allow the conclusions to be robustly supported.

Essential revisions:

1) Several aspects of the identification of P-body enriched mRNAs need to be clarified and expanded:

a) There is no clear evidence that the authors enrich for PBs, and it appears that they are really identifying mRNAs that interact with Dcp2 or Scd6 in a variation of CLIP analyses (since the RNPs are digested with T1 nuclease before they are even pelleted?), and thus they essentially analyze pelleted Dcp2 or Scd6 assemblies. This needs to be made implicit and clearly discussed since this impacts their claim that P-bodies per se are being purified. This is also important in light of the FISH analyses (see below).

b) It is unclear how enriched mRNAs are identified from the RNA-Seq data. There seems to be no comparison to the total RNA-Seq reads, and there is no control for background RNA contamination. At a minimum the methods for calling an mRNA as enriched in the RNA-Seq needs to be clarified, but the manuscript would be much stronger if the "P-body" associated mRNAs were compared to read density in total (ribo-) RNA-Seq as this should allow a more meaningful description of what mRNAs are enriched in the these fractions.

c) The authors show a principal component analysis as a measure or reproducibility. However, they should also include pair-wise correlations of their read densities and provide R squared values to assess the reproducibility

d) If the authors had a clear population of mRNAs shown to be enriched in P-bodies, one could perform computational analyses on these mRNAs to see i) how they share molecular features (length, translation efficiency, decay rates, overlap with Pat1, Lsm1 bound mRNAs identified by similar (but not identical) methods (Mitchell et al., 2013), etc.). This might provide new insights into the mechanisms of P-body formation and function.

2) A key experiment is shown in Figure 2 examining whether the identified mRNAs enrich in PBs using FISH. Again several improvements are necessary for these analyses.

a) What is the basis for using ACT1 and PGK1 as controls? How were ACT1 and PGK1 determined to not show enrichment in P-bodies from the RNA sequencing? Given the very high abundance of these mRNAs, even a low percentage of these mRNAs in P-bodies would give a high number of reads in the samples. Is this seen? This is another example wherein comparing the number of reads in the "P-body" prep as compared to the total RNA-Seq reads would be helpful.

b) It is important that the authors are explicit about numbers of mRNAs in P-bodies. It appears that an enriched mRNA (such as Bsc1 or ATP11) has approximately 15-12% of the mRNA molecules as overlapping with Dcp2. Thus, the mRNA is enriched over controls, but still a majority of molecules are outside of the P-body assembly. Is this correct? And if so, this needs to be clearly stated and discussed as this makes it more complicated to interpret how P-body localization affects function since at any one time, only a small% of the actual mRNA molecules are associated with P-bodies per se.

c) Whereas several of the identified mRNAs seem to localize to (-) glucose PBs (with the caveats above), no data for the salt-induced PBs are shown. Does this mean the enrichment protocol didn't work well for these stresses? To validate their protocol additional FISH experiments also for the salt stresses should be shown.

3) Examining how mRNA decay rates correlate with P-body localization is a good experiment but needs to be improved as follows.

a) There is a concern that the uracil chase is not effective and this is why some of the mRNAs increase during glucose deprivation relative to actin mRNA. If there is some residual labeling during the chase period, it is difficult to confidently assess how mRNA decay is changing for different classes of mRNAs. Can the authors provide clear data that the chase is robust?

b) The authors should be more cautious in the interpretation that P-body localization affects the stability/storage. In principle, these differences could be due to other regulatory circuits independent of P-bodies. As discussed above, one should be cautious about this point since the majority of mRNA molecules are not in P-bodies at any one time (although they could be cycling in and out in relevant and dynamic manner, which should be clearly stated).

c) It would be appropriate to cite the work of Nissan's group who has recently argued that P-bodies can protect some mRNAs during stress (Huch and Nissan, 2017).

4) The experiments showing Puf5 and the 3' UTR of ATP11 can both affect its concentration in P-bodies and its decay rate raise the possibility that P-body association of this mRNA increases its stability. However, the data could also be interpreted as ATP11 is targeted to P-bodies, and if bound by Puf5, decapping/deadenylation is slowed such that the dwell time within P-bodies is longer and hence more mRNAs at steady state are associated with P-bodies. It would strengthen the manuscript (and be appropriate for publication in a high profile venue like *eLife*) to resolve this issue particularly also in the light of the observation that the Puf5-independent PB-associated BSC1 mRNA is stabilized in the absence of Puf5. Whether Puf5 affects P-body targeting per se, or decapping within it, could be assessed by measuring the association of ATP11 with P-bodies in puf5∆, dcp1∆, puf5∆ dcp1∆ strains, where the affect of Puf5 on ATP11 in the puf5∆ dcp1∆ should reveal whether it affect targeting of mRNAs into P-bodies or their decay rate once there.

Also, is tethering of Puf5 (via MS2 loops or similar) sufficient for PB localization?

[Editors' note: further revisions were requested prior to acceptance, as described below.]

Thank you for submitting your article "Context-dependent deposition and regulation of mRNAs in P-bodies" for consideration by *eLife*. Your article has been reviewed by two peer reviewers, one of whom is a member of our Board of Reviewing Editors, and the evaluation has been overseen by James Manley as the Senior Editor. The following individual involved in review of your submission has agreed to reveal his identity: Roy Parker (Reviewer #2).

The reviewers have discussed the reviews with one another and the Reviewing Editor has drafted this decision to help you prepare a revised submission.

Summary and Essential revisions:

During the first round of revision, the authors were able to address several concerns that were raised previously, which has improved the manuscript. However, there are some remaining issues that require attention.

1) The reviewers remain unconvinced that the authors specifically purify mRNAs 'enriched in P-bodies', both because of the nature of RNAse treatment before purification, and because the analysis to identify enriched mRNAs relies on comparing the mRNAs associated with Dcp2 or Scd6 without stress, to those associated with these proteins during stress. Thus, the impact is really to identify mRNAs associated with these proteins under different stresses. This is a useful contribution but this point should be clear and be reflected in the wording. For example, reviewer 1 suggests that the authors use wordings such as mRNAs "associated with PB components" instead of "enriched in PBs" throughout the text.

2) The manner by which the enriched mRNAs identified should be shown in a logical flow chart (could be in supplemental). Reviewer 2 asks for this since he is still not sure how the analysis was done and the high similarity between RNA-Seq data sets under all conditions makes him unclear about the statistics. Since all of the manuscript flows from this analysis, it needs to be clear how it was done and what statistical tests were used.

3) The Figure 1 flow chart does not match the written description in the Materials and methods. This adds to the confusion as to how the experiment was done and needs to be clarified.

---

## [Author Response]

Essential revisions:1) Several aspects of the identification of P-body enriched mRNAs need to be clarified and expanded:a) There is no clear evidence that the authors enrich for PBs, and it appears that they are really identifying mRNAs that interact with Dcp2 or Scd6 in a variation of CLIP analyses (since the RNPs are digested with T1 nuclease before they are even pelleted?), and thus they essentially analyze pelleted Dcp2 or Scd6 assemblies. This needs to be made implicit and clearly discussed since this impacts their claim that P-bodies per se are being purified. This is also important in light of the FISH analyses (see below).

We agree with the reviewers that we are, strictly speaking, enriching for mRNAs that can be crosslinked to Dcp2p and Scd6p containing assemblies and that are membrane associated. Our previous data indicated that most P-bodies in yeast are ER-associated (Kilchert et al., 2010, Weidner et al., 2014). Initially, we tried the crosslinking approach using just cytosol. To our surprise, we were unable to crosslink any proteinous P-body components to any appreciable extent to Dcp2p tagged with HBH. In contrast, using the P20 membrane fraction allowed the pull-down of other P-body components (Weidner et al., 2014). This does not formerly exclude the existence of non-membrane associated P-bodies as the free Dcp2p or Scd6p may simply bind much better to the resin than the same protein present in a tightly assembled RNP. Nevertheless, we showed previously that with the cross-linking affinity chromatography approach, we can pull down assemblies that contain P-body components. We agree with the reviewers that we cannot exclude the presence of other assemblies in the pulldown. Therefore, we are now more cautious and replaced ‘purified’ with ‘enriched’.

We modified the Results section and inserted the following statements:

“Strictly speaking we are enriching mRNAs that can be crosslinked to Dcp2p or Scd6p or their interaction partners under stress conditions. Given that we identified P-body components previously using this method (Weidner et al., 2014), and that the Scd6p experiment clustered well with the ones performed with Dcp2p, makes it likely that the RNAs, we identified are present in P-bodies.”

“To ensure the specificity of the mRNAs associated with P-body components, we performed RNA-Seq experiments on the total RNA content under control as well as different stress conditions (Figure 1—figure supplement 1, Supplementary file 3). mRNAs that were upregulated upon any of the stresses, as determined by total RNA-Seq, were generally not enriched in the corresponding P-body fraction with less than 15% overlap between the RNAs generally upregulated in stress response and the RNAs pulled down by P-body components (Figure 1). Moreover, from the glucose-starvation enriched mRNA pool, polysome-associated mRNAs identified under the same stress (Arribere et al., 2011) were eliminated.”

In addition, we modified the Discussion, Figure 1 and the Figure legend of Figure 1 at several places to indicate that with the pull-down we detect RNAs that are associated with P-body components at membranes.

b) It is unclear how enriched mRNAs are identified from the RNA-Seq data. There seems to be no comparison to the total RNA-Seq reads, and there is no control for background RNA contamination. At a minimum the methods for calling an mRNA as enriched in the RNA-Seq needs to be clarified, but the manuscript would be much stronger if the "P-body" associated mRNAs were compared to read density in total (ribo-) RNA-Seq as this should allow a more meaningful description of what mRNAs are enriched in the these fractions.

We apologize, if we were clear enough in Material and Methods about how we determined enriched mRNAs. We had four conditions in each experiment: 1 unstressed control and 3 stress conditions. mRNAs that were crosslinked to Dcp2 or Scd6 under stress but not in the unstressed control were assumed to be enriched. We identified significantly (*p*<0.05) upregulated mRNAs exclusive for each stress condition by testing each individual stress condition against the wild type condition and removing those mRNAs that were identified as common hits when testing the joint set of all three stress conditions against unstressed control. For glucose depletion stress, we additionally excluded genes previously shown to be significantly enriched in polysomes (Arribere et al., 2011) for the same stress.

We additionally performed total RNA-Seq of untreated, glucose-starved, NaCl^-^ and CaCl_2_-stressed *S. cerevisiae* cultures (3 independent samples per condition), and identified significant (p<0.05) upregulated mRNAs for each condition by following the same procedure described for P-body associated mRNAs. In all cases, we find that less than 15% of the RNAs that are enriched under any stress overlap between the total and the mRNAs, we identified by cCLAP. Thus, we are able to enrich P-body associated mRNAs with our method – or as discussed above mRNAs that can be crosslinked to Scd6, Dcp2 or their interactors under stress. We provide the principle component analysis (Figure 1—figure supplement 1), pair-wise correlations (Figure 1—figure supplement 1) and an excel file listing the enriched RNAs from the newly included total RNA-Seq as supplementary material. The new sequencing data were uploaded under the same GEO accession number than the P-body RNA-Seq data.

c) The authors show a principal component analysis as a measure or reproducibility. However, they should also include pair-wise correlations of their read densities and provide R squared values to assess the reproducibility

The correlation analysis and the R^2^ values are now provided in Figure 1—figure supplement 1.

d) If the authors had a clear population of mRNAs shown to be enriched in P-bodies, one could perform computational analyses on these mRNAs to see i) how they share molecular features (length, translation efficiency, decay rates, overlap with Pat1, Lsm1 bound mRNAs identified by similar (but not identical) methods (Mitchell et al., 2013), etc.). This might provide new insights into the mechanisms of P-body formation and function.

We searched the lists of mRNAs UV-crosslinked to Dhh1p, Sbp1p, Pat1p and Lsm1p from Mitchell et al., 2013 for the Group I and Group II candidates that we had identified. Of the three mRNAs degraded in P-bodies, only *TPI1* was found, which is a highly abundant mRNA. *BSC1* and *RLM1* were not present. Of the group II (stabilized mRNAs), we found *ATP11, ILM1* and *MRPL38*. Even though the sample size is limited, this finding would suggest that the decayed RNAs polyA-tail shortening occurs early on and hence such RNAs would not be captured by the oligo-dT enrichment. Our unbiased approach would still be able to pick up such RNAs. In contrast, the stabilized mRNAs, which should still contain a polyA-tail, would be detectable by both methods.

Comparing more globally our hit lists with those from Mitchell et al., 2013, we found that there was surprisingly little overlap. There are a number of reasons, why this might be the case. First of all, the stress applied in both cases was not the same. We used specifically glucose starvation, meaning we used full (YP) medium, from which glucose was omitted, and for the different salt stresses, salts were added to the given concentration. Mitchell et al., 2013 stressed their cells in PBS, depleting cells not only from the carbon source, but also the nitrogen source, amino acids, components essential for nucleotide synthesis etc and a condition that might also yield hypotonic stress. Thus, Mitchell et al. applied a more complex stress, which was perfectly suitable for their analysis to identify the sites of interaction of P-body components with client RNA. Our question was different in that we wanted to determine, which mRNAs are deposited into P-bodies under a specific stress. As such, we had to make sure that we only changed one parameter and avoided complex stresses. Second, as outlined above, oligo-dT will enrich for mRNAs still having an intact polyA-tail, which is part of the method used in the Parker lab. We opted for a more unbiased approach to allow for the detection of mRNAs that have lost most or all of their polyA-tail. Finally, the protocols differ in quite a few additional ways, most notably, the cross-linking method (UV cross-linking vs 2 min formaldehyde cross-linking, and our method not requiring any harvesting step before cross-linking as we cross-linked and quenched in the culture medium), different starting material (whole cell lysate vs membrane fractions), different proteins used for pulldowns, and depletion of ribosomes and ribosomal RNAs in our protocol. The differences in the selection of the RNA and in the protocols may also contribute to the finding that the population of the mRNAs that we identified under all three stresses to be associated with P-body components is more similar than under the individual stresses but still far from being a good overlap with the hits from the Mitchell paper (less than 20%). It is also worth to point out that the bioinformatic analyses performed are different and may also contribute to the difference in the hit-lists.

We compared the length of genes, as obtained from the gtf file, associated with P-body components versus the ones generally upregulated under a particular stress and found that mRNAs associated with P-body components are shorter under glucose starvation and longer under both hyperosmotic stress conditions, when compared to total RNA levels. These results indicate that the gene length might provide a bias towards P-body component association. This analysis has been included into the manuscript.

Since there was little overlap between the RNAs from Mitchell et al., 2013 and our dataset, which might be due to the difference in stress application and the experimental procedure, we did not discuss these discrepancies in the manuscript. The Mitchell paper provides an extremely important study about where particular P-body components bind (something we have not addressed at all, as our focus was more on the different RNA species).

2) A key experiment is shown in Figure 2 examining whether the identified mRNAs enrich in PBs using FISH. Again several improvements are necessary for these analyses.a) What is the basis for using ACT1 and PGK1 as controls? How were ACT1 and PGK1 determined to not show enrichment in P-bodies from the RNA sequencing? Given the very high abundance of these mRNAs, even a low percentage of these mRNAs in P-bodies would give a high number of reads in the samples. Is this seen? This is another example wherein comparing the number of reads in the "P-body" prep as compared to the total RNA-Seq reads would be helpful.

*ACT1* and *PGK1* are very commonly used mRNAs for normalization purposes for qRT-PCR experiments. To ensure that *ACT1* and *PGK1* could be used for normalization of our qRT-PCR experiments, we tested initially the stability of *ACT1, PGK1* and *RPL37b* under glucose starvation using spiked-in humanized *Renilla* luciferase RNA for normalization Figure 3—figure supplement 1. *ACT1* and *PGK1* were then chosen because those mRNAs relatively stable, in particular when compared to *RPL37b*. Now, also the total RNA-Seq confirmed that *PGK1* and *ACT1* do not significantly increase under the stresses applied. Moreover, neither *ACT1* nor *PGK1* were found to be enriched in P-bodies to any significant extent, when compared to the total mRNA and thus, independently confirming the suitability of *ACT1* and *PGK1* for normalization of our qRT-PCRs. We do find *ACT1* and *PGK1* in the P-body associated mRNAs fractions. However, similar levels were also found in the unstressed control, most likely due to the high abundance of these mRNAs, as the reviewers already suggested. Hence, they were not enriched under any condition and thus not taken into account.

b) It is important that the authors are explicit about numbers of mRNAs in P-bodies. It appears that an enriched mRNA (such as Bsc1 or ATP11) has approximately 15-12% of the mRNA molecules as overlapping with Dcp2. Thus, the mRNA is enriched over controls, but still a majority of molecules are outside of the P-body assembly. Is this correct? And if so, this needs to be clearly stated and discussed as this makes it more complicated to interpret how P-body localization affects function since at any one time, only a small% of the actual mRNA molecules are associated with P-bodies per se.

We agree with the reviewers that only a fraction of the mRNA is localized to the P-bodies. There are several possibilities to explain these findings. The most plausible one is that we perform the FISH-IF after only 10’ of stress induction. This time point was chosen because stress granules are absent, in all stresses at after 10’, facilitating the analysis. However, this comes with the caveat that probably not all mRNAs have yet been targeted to P-bodies. Under glucose starvation, P-bodies grow over time, suggesting that more mRNAs are also deposited there. Another potential caveat of our approach is that we are likely underestimating the number of mRNA molecules in P-bodies. While mRNAs in the cytoplasm are easily detected by long probes (even up to 1,000 bp), those fail to detect mRNAs in P-bodies. We use smaller probes and can detect signals in P-bodies, yet, we have no idea about the efficiency of the labelling. We did not hide the fact that only a fraction of any particular mRNA is associated with P-bodies. In fact, when setting the threshold of 1.5 to call an mRNA enriched, we discussed the points above and some additional ones (subsection “The nature of P-body sequestered RNAs is stress-dependent”, paragraph three).

We do not think that only a small fraction of mRNA might be present in P-bodies. Clearly our FISH-IF method is not a single molecule FISH method. On top of the consideration above, we use an enzymatic reaction involving tyramide for the detection of the mRNA. When we observe mRNA co-localization within P-bodies, it does not allow us to make any statement about how many mRNA molecules are present in the P-body at that particular time; this could be one, this could be ten. This was an additional reason, why we did not talk about percentage of mRNA signal overlapping with P-bodies but rather about enrichment in P-bodies. We now mention that we use a non-quantitative method to visualize the mRNA in terms of number of molecules in the P-body.

c) Whereas several of the identified mRNAs seem to localize to (-) glucose PBs (with the caveats above), no data for the salt-induced PBs are shown. Does this mean the enrichment protocol didn't work well for these stresses? To validate their protocol additional FISH experiments also for the salt stresses should be shown.

We agree with the reviewers and include now also data on P-body-localized mRNA under CaCl_2_ and NaCl stresses. The data are part of Figure 2—figure supplement 1.

3) Examining how mRNA decay rates correlate with P-body localization is a good experiment but needs to be improved as follows.a) There is a concern that the uracil chase is not effective and this is why some of the mRNAs increase during glucose deprivation relative to actin mRNA. If there is some residual labeling during the chase period, it is difficult to confidently assess how mRNA decay is changing for different classes of mRNAs. Can the authors provide clear data that the chase is robust?

We used the same uracil concentration (20 mM) for the chase as has been used by other groups previously. To exclude that the uracil concentration was too low to efficiently compete out 4-Thiouracil, we sought to add the highest possible concentration of uracil, which is 32 mM due to solubility issues. Using 32 mM uracil in the chase, did not change our results on mRNA stability and decay.

In the original study, we did not want to use transcriptional inhibitors because adding them is obviously also stressful for the cell, and we would not be able to distinguish between stability/decay induced by glucose starvation or transcriptional shut off. However, we assume that using transcriptional inhibitors to provide a second, independent confirmatory line to examine RNA decay rates should be acceptable. We therefore treated cells with 1,10 phenanthroline during glucose starvation and determined decay rates. As for the 4TU experiments, Group II mRNAs were stabilized (except for *ILM1*, which was somewhat less stable, presumably due to the transcription block stress), while Group I mRNAs were still decayed.

Taken together, we are confident that the uracil chase is indeed efficient and robust. These data are now provided in Figure 3—figure supplement 1.

b) The authors should be more cautious in the interpretation that P-body localization affects the stability/storage. In principle, these differences could be due to other regulatory circuits independent of P-bodies. As discussed above, one should be cautious about this point since the majority of mRNA molecules are not in P-bodies at any one time (although they could be cycling in and out in relevant and dynamic manner, which should be clearly stated).

We added a sentence to this effect in the Results section and in the Discussion section.

c) It would be appropriate to cite the work of Nissan's group who has recently argued that P-bodies can protect some mRNAs during stress (Huch and Nissan, 2017).

We agree with the reviewers and have included the Huch and Nissan, 2017 reference into the manuscript.

4) The experiments showing Puf5 and the 3' UTR of ATP11 can both affect its concentration in P-bodies and its decay rate raise the possibility that P-body association of this mRNA increases its stability. However, the data could also be interpreted as ATP11 is targeted to P-bodies, and if bound by Puf5, decapping/deadenylation is slowed such that the dwell time within P-bodies is longer and hence more mRNAs at steady state are associated with P-bodies. It would strengthen the manuscript (and be appropriate for publication in a high profile venue like eLife) to resolve this issue particularly also in the light of the observation that the Puf5-independent PB-associated BSC1 mRNA is stabilized in the absence of Puf5. Whether Puf5 affects P-body targeting per se, or decapping within it, could be assessed by measuring the association of ATP11 with P-bodies in puf5∆, dcp1∆, puf5∆ dcp1∆ strains, where the affect of Puf5 on ATP11 in the puf5∆ dcp1∆ should reveal whether it affect targeting of mRNAs into P-bodies or their decay rate once there.

Thank you for the great suggestion! We generated a *∆puf5 ∆dcp1* strain and assessed the *ATP11* mRNA stability and the level of co-localization of *ATP11* mRNA with the P-body marker Dcp2p-GFP in this strain. In the double mutant *ATP11* mRNA was decay with similar kinetics than in the *∆puf5* mutant. Likewise, we did not detect any accumulation of *ATP11* mRNA in P-bodies in the *∆puf5 ∆dcp1* cells. Therefore, it seems more likely that Puf5p is involved into the targeting rather than modulating the decay rate. As you will see below, in parallel to targeting Puf5p to P-bodies, we also made the same construct for Puf3p. In the latter case, we observed *ATP11* mRNA decay, suggesting that Puf5p might be involved in targeting and Puf3p in the stability of *ATP11* mRNA to/in P-bodies.

Also, is tethering of Puf5 (via MS2 loops or similar) sufficient for PB localization?

This is an interesting idea! We constructed a GFP-U1A binding protein-Puf5 fusion protein and used two independent constructs with U1A loops for targeting to P-bodies. In both cases, in about 40-50% of cells expressing both constructs well, we observed Puf5p co-localization with Dcp2p-mCherry. This co-localization had no effect on the stability of *ATP11* mRNA. It is worthwhile to note, however, that most of the GFP-U1A binding protein-Puf5 fusion protein remained in the cytoplasm and also accumulated on vacuoles, complicating the interpretation of the RNA decay data and potential FISH-IF (P-body maker and *ATP11* mRNA). We would need to be able to detect GFP, mCherry and the mRNA simultaneously. Unfortunately, we are currently not equipped to do triple colour FISH-IF on P-body-associated mRNAs.

In parallel, we had also generated a fusion with Puf3p. Unlike Puf5p, Puf3p was targeted to the P-bodies very efficiently, indicating that our targeting-to-P-bodies approach worked in principle. In fact, targeting Puf3p to P-bodies caused the decay of *ATP11* mRNA, confirming a role for Puf3p in *ATP11* stability as suggested by Miller et al., 2014.

We added the experiments to manuscript and updated our model accordingly. We now also speculate about a role Puf3p in *ATP11* mRNA stability.

[Editors' note: further revisions were requested prior to acceptance, as described below.]

During the first round of revision, the authors were able to address several concerns that were raised previously, which has improved the manuscript. However, there are some remaining issues that require attention.1) The reviewers remain unconvinced that the authors specifically purify mRNAs 'enriched in P-bodies', both because of the nature of RNAse treatment before purification, and because the analysis to identify enriched mRNAs relies on comparing the mRNAs associated with Dcp2 or Scd6 without stress, to those associated with these proteins during stress. Thus, the impact is really to identify mRNAs associated with these proteins under different stresses. This is a useful contribution but this point should be clear and be reflected in the wording. For example, reviewer 1 suggests that the authors use wordings such as mRNAs "associated with PB components" instead of "enriched in PBs" throughout the text.

As requested by the reviewers, we do not state in the manuscript that we purify mRNA enriched in P-bodies with our cCLAP method, only mRNA that is associated with P-body components.

2) The manner by which the enriched mRNAs identified should be shown in a logical flow chart (could be in supplemental). Reviewer 2 asks for this since he is still not sure how the analysis was done and the high similarity between RNA-Seq data sets under all conditions makes him unclear about the statistics. Since all of the manuscript flows from this analysis, it needs to be clear how it was done and what statistical tests were used.

We provide the logical flow chart/pipeline as requested by reviewer 2, reflecting the description in Material and Methods in the hope that is clarifies any misunderstanding. This chart is now presented in Figure 1—figure supplement 2.

Just for the record, we respectfully disagree that all of the manuscript flows from this analysis because we independently confirmed our hits by FISH-IF. In addition, independent of the wealth of information and resource the RNAseq data provide, the description of the fates of Group I and Group II mRNAs in P-bodies, the involvement of Puf5 in this fate determination, and the importance of ATP11 mRNA levels for chronological lifespan extension could stand on their own.

3) The Figure 1 flow chart does not match the written description in the Materials and methods. This adds to the confusion as to how the experiment was done and needs to be clarified.

We modified the scheme in Figure 1 to indicate the RNase treatments and corrected the centrifugal force applied to the samples. We hope that this clarifies the experimental outline.